# FERA: UNCERTAINTY-AWARE FEDERATED REASONING FOR LARGE LANGUAGE MODELS

## ABSTRACT

Reasoning capabilities in large language models (LLMs) are critical for advancing beyond pattern recognition toward systematic problem-solving, logical inference, and reliable decision-making. Most reasoning LLM (rLLM) approaches rely on fine-tuning but are constrained by privacy barriers that prevent centralizing domain-specific reasoning data. Federated methods enable privacy-preserving collaboration but come with prohibitive computational and communication costs. To address these challenges, we propose the Uncertainty-aware Federated Reasoning (FERA) framework that eliminates costly training and reduces communication overhead while preserving strong reasoning performance under heterogeneous data. FERA introduces *Uncertainty-Aware Aggregation*, using iterative server–client collaboration where uncertainty estimates are progressively refined across communication rounds to guide response generation. We establish theoretical convergence guarantees and validate our framework through extensive experiments, demonstrating that FERA achieves state-of-the-art reasoning accuracy with substantially greater efficiency than existing approaches.

## 1 INTRODUCTION

Large language models (LLMs) have advanced rapidly in recent years, fundamentally reshaping natural language processing by moving beyond simple pattern recognition toward systematic reasoning. An important milestone in this evolution is the emergence of reasoning ability, defined as the process of answering questions that require complex, multi-step generation with intermediate steps (Liu et al., 2024; Achiam et al., 2023; Wei et al., 2022; Kojima et al., 2022). Reasoning Large language models (rLLMs) are thus designed to handle challenging tasks such as solving puzzles, tackling advanced mathematical problems, and performing sophisticated coding (Jiang et al., 2025; Xu et al., 2025).

The standard methodology for creating effective rLLMs involves fine-tuning models with large, centralized reasoning datasets to learn systematic reasoning patterns across diverse problem domains (Li et al., 2025; Zhu et al., 2025; Yu et al., 2025). However, the requirement to centralize data conflicts with modern data governance practices and privacy concerns. The most valuable reasoning data for training rLLMs—domain-specific datasets containing rich, contextual reasoning examples—is inherently distributed across different organizations and institutions, each constrained by privacy regulations that prohibit data sharing. As a result, the approach that drives rLLM development proves difficult to apply in practical deployment settings (Wei et al., 2025).

*Federated learning* has emerged as a promising paradigm to resolve this centralization-privacy dilemma by enabling collaborative model development across distributed data sources while maintaining data locality (Ye et al., 2024; Chen et al., 2024). This approach naturally gives rise to Federated reasoning (Fed-reasoning), where multiple clients can jointly enhance reasoning capabilities through decentralized knowledge sharing without compromising data sovereignty. Current Fed-reasoning research predominantly follows training-based methodologies, wherein participating clients train local model replicas on their reasoning datasets and share only aggregated parameter updates with a central coordinator (Wang et al., 2024c; Wu et al., 2024a; Ma et al., 2023). However, existing training-based Fed-reasoning approaches suffer from critical limitations that severely constrain their practical applicability (Yan et al., 2025; Shu et al., 2024). First, these methods demand extensive local model training across all distributed clients, imposing substantial computational costs

Figure 1: Overview of the FERA framework workflow. Clients leverage the global context distributed by the server to relabel their local datasets and iteratively refine their contextual representations. The server aggregates the updated client responses to construct a new global context, enabling privacy-preserving collaboration across heterogeneous client data. The core component, **UA-SCA**, performs uncertainty-aware aggregation of client responses. Detailed explanations of these components are provided in Section 3.1.

at each participating node—creating prohibitive barriers for resource-constrained organizations (Liu et al., 2023; Che et al., 2023). Second, the iterative nature of federated training generates amplified communication overhead, as large model parameter updates must be transmitted across potentially bandwidth-limited networks through multiple training rounds, creating scalability bottlenecks that undermine deployment feasibility in large-scale federated environments (Wang et al., 2024d; Liu et al., 2025).

To address these computational and communication challenges, an alternative line of research has explored *parameter-free* approaches that eliminate training requirements entirely. These methods reduce communication overhead by transmitting only contextual information rather than model parameters, enabling clients to leverage their existing local LLMs for collaborative reasoning (Chen et al., 2025; Wang et al., 2025a; Wu et al., 2024c; Du et al., 2023). While parameter-free approach provides clear advantages in both computation and communication, they introduce fundamental performance trade-offs: high sensitivity to client data heterogeneity and degraded performance on complex reasoning tasks that require sophisticated cross-client knowledge integration (Jimenez G et al., 2024; Chen & Vikalo, 2024; Mendieta et al., 2022). This landscape reveals a key gap in federated reasoning: training-based methods achieve strong performance but are computationally costly, whereas parameter-free approaches are efficient yet falter under data heterogeneity and complex reasoning. This dilemma motivates the key question we address:

> *How can we design parameter-free methods to handle complex reasoning tasks while maintaining robustness under heterogeneous data.?*

To address these limitations, we propose the **Uncertainty-Aware Federated Reasoning (FERA)** framework. Our key contributions are summarized as follows:

- We introduce FERA, which operates through iterative server–client collaboration, as illustrated in Figure 1. In each communication round, the server distributes questions to clients with private, high-quality reasoning data. The clients then generate reasoning steps for these questions based on their local data, and the server subsequently aggregates the collected reasoning steps. This principled design enables FERA to achieve robust reasoning performance while remaining effective under heterogeneous client data.
- To address the heterogeneity of local reasoning data across clients, we propose an **Uncertainty-Aware Aggregation** method that assigns weights to client contributions based on their estimated uncertainty. Intuitively, higher uncertainty suggests weaker relevance of the client's data to the server's question, thus warranting a lower weight in aggregation. We further develop **Uncertainty-Aware Self-Critique Aggregation (UA-SCA)**, which estimates uncertainty via self-critique (Gao et al., 2024). This differs from existing uncertainty-aware federated learning frameworks (Zhang & Yu, 2025), which rely solely on client-level uncertainty rather than critique of individual questions.
- On the theoretical side, we analyze a simplified *linear self-attention* model (Zhang et al., 2023) to formalize the benefits of FERA's uncertainty-aware framework. Our results show that the aggre-

gated answers produced by FERA converge to the ground-truth answers of the server's questions. More importantly, incorporating uncertainty-aware weights provably accelerates the convergence rate, thereby validating the design of our aggregation strategy.

- We conduct extensive experiments across diverse task categories, ranging from general reasoning tasks to specialized mathematical problems. The results demonstrate that FERA consistently outperforms strong baselines, and ablation studies further confirm the contribution of each individual component.

## 2 RELATED WORK

Recent advances in FL and LLMs have converged around uncertainty as a unifying principle for addressing key challenges (see Appendix B for detailed discussion). In federated reasoning, approaches divide into training-driven methods that update parameters efficiently like FedKSeed (Qin et al., 2023) and FLoRA (Wang et al., 2024c), and lightweight training-free methods that rely on prompt engineering and in-context learning without parameter updates, but struggle with data heterogeneity and complex reasoning tasks. Uncertainty quantification has proven crucial for FL systems in handling data heterogeneity (Koutsoubis et al., 2025), improving model transparency (Zhang & Yu, 2025), and bridging architectural differences (Wang et al., 2024a; Zhang et al., 2024b). Similarly, uncertainty in LLMs enables decomposition of aleatoric and epistemic components (Hou et al., 2023), guides active learning (Huang et al., 2024), and supports adaptive decision-making through self-correction or abstention (Wang et al., 2025b; Yang et al., 2023). These parallel developments motivate our unified framework, FERA, which leverages uncertainty to enhance federated reasoning without requiring parameter updates.

## 3 UNCERTAINTY-AWARE FEDERATED REASONING

We propose the Uncertainty-aware Federated Reasoning (**FERA**) framework in this section, beginning with the federated learning setup (Wang et al., 2025a). A central server coordinates $L$ clients, where each client $i$ holds a private dataset $D^i = \{(q_n^i, s_{\{1:T\},n}^i, a_n^i)\}_{n=1}^N$. Here $q_n^i$ is a query (question), $s_{\{1:T\},n}^i$ the sequence of $T$ reasoning steps, and $a_n^i$ the final answer. The server maintains a query set $Q = \{(q_m, s_{\{1:T\},m}, a_m)\}_{m=1}^M$. For each new query $q_m$, client $i$ selects demonstrations from $D^i$, appends $q_m$, and prompts its local LLM, named $\text{LLM}^i$, to generate a response. Building on this setup, FERA employs an iterative refinement workflow that updates answers using feedback from clients' local data and uncertainty estimates. The workflow, summarized in Algorithm 1, proceeds as follows:

**Step 1 (Server Distribution).**    At round $k$, the server distributes the query dataset $Q_k$ to all clients.

**Step 2 (Local Refinement).**    Next, client $i$ first selects a subset of demonstrations $\mathcal{S}_{k,Q}^i \subseteq Q_k$ from the server's query set. The selected demonstrations are incorporated into prompts to guide the local model $\text{LLM}^i$ to generate refined reasoning–answer pairs through $(s_{\{1:T\},k,n}^i, a_{k,n}^i) \sim \text{LLM}^i(s_{\{1:T\}}, a \mid q_n^i, \mathcal{S}_{k,Q}^i)$, yielding the enhanced dataset $D_k^i = D^i \cup \{(q_n^i, s_{\{1:T\},k,n}^i, a_{k,n}^i)\}_{n=1}^N$.

Specifically, the demonstration selection uses the Maximal Marginal Relevance (MMR) strategy (Carbonell & Goldstein, 1998):

$$\mathcal{S}_{k,Q}^i = \arg\max_{\mathcal{S} \subseteq Q_k} \left[ \frac{1}{|\mathcal{S}|} \sum_{(q', s'_{1:T}, a') \in \mathcal{S}} \text{Sim}\big((q_n^i, s_{1:T,k-1,n}^i, a_{k-1,n}^i), (q', s'_{1:T}, a')\big) - \lambda \cdot \text{Div}(\mathcal{S}) \right], \quad (3.1)$$

where Sim is the similarity between the embedding of the query example and the other examples in the dataset, $\lambda > 0$ is the regularization parameter, and $\text{Div}(\mathcal{S})$ is a diversity measure of set $\mathcal{S}$. More details about MMR are deferred to Appendix E.5.

**Step 3 (Client Labeling).**    Next, client $i$ uses its updated dataset $D_k^i$ to generate predictions $\{a_{k+1,m}^i\}_{m=1}^M$ for the server queries $\{q_m\}_{m=1}^M$. For each query, the client constructs a demon-

---

**Algorithm 1** Uncertainty-Aware Federated Reasoning

---

**Require:** $L$ clients, each with local dataset $D^i = \{(q_n^i, s_{\{1:T\},n}^i, a_n^i)\}_{n=1}^N$ and local model LLM$^i$.
Server holds initial query set $\{q_m\}_{m=1}^M$.
1: Initialize server query set $Q_1 = \{(q_m, s_{\{1:T\},1,m}, a_{1,m})\}_{m=1}^M$.
2: **for** each round $k = 1, \ldots, K$ **do**
3:     *Step 1:* Server distributes $Q_k$ to all clients.
4:     **for** each client $i = 1, \ldots, L$ **do**
5:         *Step 2:* Refine local reasoning–answer pairs using demonstrations from $Q_k$. Form the enhanced dataset:
$$D_k^i \leftarrow D^i \cup \{(q_n^i, s_{\{1:T\},k,n}'^i, a_{k,n}'^i)\}_{n=1}^N.$$
6:         *Step 3:* Predict reasoning–answer pairs for server queries using demonstrations from $D_k^i$. Compute uncertainty scores from the predictive distribution of LLM$^i$, and return to the server:
$$\{(s_{\{1:T\},k+1,m}^i, a_{k+1,m}^i, u_{k+1,m}^i)\}_{m=1}^M.$$
7:     **end for**
8:     *Step 4:* The server aggregates client responses using an uncertainty-aware scheme; see Section 3.1 for aggregation procedures under different algorithmic settings.
        **Update**
$$Q_{k+1} = \{(q_m, s_{\{1:T\},k+1,m}, a_{k+1,m})\}_{m=1}^M.$$
9: **end for**
**Ensure:** Final predictions $\{(q_m, s_{\{1:T\},K+1,m}, a_{K+1,m})\}_{m=1}^M$.

---

stration set $\mathcal{S}_{k,D}^i \subseteq D_k^i$ and uses it to guide the local model LLM$^i$ in generating updated responses for the server queries: $(s_{\{1:T\},k+1,m}^i, a_{k+1,m}^i) \sim$ LLM$^i(s_{\{1:T\}}, a \mid q_m, \mathcal{S}_{k,D}^i)$.

The demonstration set $\mathcal{S}_{k,D}^i$ is also selected using the MMR strategy to balance relevance to the current query and diversity within the selected examples:

$$\mathcal{S}_{k,D}^i = \arg \max_{\mathcal{S} \subseteq D_k^i} \left[ \frac{1}{|\mathcal{S}|} \sum_{(q', s_{1:T}', a') \in \mathcal{S}} \text{Sim}\big((q_m, s_{1:T,k,m}, a_{k,m}), (q', s_{1:T}', a')\big) - \lambda \cdot \text{Div}(\mathcal{S}) \right]. \quad (3.2)$$

Here Sim, $\lambda$, and Div are the same functions and parameter as defined in Step 2. After generating the reasoning steps and answers for query $q_m$, FERA computes the uncertainty $u_{k+1,m}^i$ based on the client LLM output logits of $(s_{\{1:T\},k+1,m}^i, a_{k+1,m}^i)$, following existing approaches to uncertainty estimation (Duan et al., 2023; Farquhar et al., 2024; Zhang et al., 2025). More details on uncertainty estimation are provided in Appendix E.5.

**Step 4 (Server Update).** The server aggregates client predictions via an uncertainty-aware scheme to update the dataset $Q_{k+1} = \{(q_m, s_{\{1:T\},k+1,m}, a_{k+1,m})\}_{m=1}^M$.

We have several remarks for Algorithm 1.

**Remark 3.1.** A recent work has explored incorporating federated learning with in-context learning for QA tasks (Wang et al., 2025a). In contrast, FERA targets more challenging reasoning tasks, and its algorithmic design is fundamentally different.

**Remark 3.2.** In general, FERA requires the server only to aggregate the reasoning steps and answers provided by clients, without additional computations, similar to Wang et al. (2025a). However, as discussed in Step 4, FERA must adopt a specific aggregation scheme for reasoning steps. In practice, this requires the server to be equipped with its own LLM to perform the aggregation.

**Remark 3.3.** FERA only requires clients to transmit reasoning steps and answers for the server's questions, without uploading their private datasets. This design preserves the privacy requirements of federated learning.

Overall, FERA belongs to the class of parameter-free federated reasoning frameworks (Wang et al., 2025a; Chen et al., 2025). To ensure strong performance on complex reasoning tasks and robustness to data heterogeneity across clients, FERA incorporates a key component: the *Uncertainty-Aware Aggregation* mechanism, which is described in detail in the following section.

---

**Algorithm 2** Uncertainty-Aware Self-Critique Aggregation

---

**Require:** Query index $m$, round $k$, client submissions $\{(s^i_{\{1:T\},k,m}, a^i_{k,m}, u^i_{k,m})\}^L_{i=1}$, the server
    LLM denoted by LLM$^S$.
**Ensure:** Aggregated reasoning–answer pair $(s^\star, a^\star)$.
  1: Partition $\{(s^i_{\{1:T\},k,m}, a^i_{k,m})\}^L_{i=1}$ into groups $\mathcal{G} = \{G\}$ based on $a_m$.
  2: **for** each group $G \in \mathcal{G}$ **do**
  3:    $S_G \leftarrow \mathsf{Summarize}(\{s^i_{\{1:T\},k,m} : (s^i_{\{1:T\},k,m}, a^i_{k,m}) \in G\}; \text{LLM}^S)$.
  4: **end for**
  5: **for** $i = 1, \ldots, L$ **do**
  6:    Denote $G(i) \in \mathcal{G}$ as the group which $(s^i_{\{1:T\},k,m}, a^i_{k,m})$ belongs to.
  7:    $(\widehat{s}^i_{\{1:T\},k,m}, \widehat{a}^i_{k,m}) \leftarrow \mathsf{SelfCritique}((s^i_{\{1:T\},k,m}, a^i_{k,m}), \{S_G : G \neq G(i)\}; \text{LLM}^S)$.
  8: **end for**
  9: Calculate weights $\{w_i\}^L_{i=1}$ based on 3.3.
10: $(s_{\{1:T\},k+1,m}, a_{\{1:T\},k+1,m}) \leftarrow \mathsf{Aggregate}(\{(\widehat{s}^i_{\{1:T\},k,m}, \widehat{a}^i_{k,m}, w_i)\}^L_{i=1}; \text{LLM}^S)$.
11: **return** $(s_{\{1:T\},k+1,m}, a_{\{1:T\},k+1,m})$.

---

## 3.1 UNCERTAINTY-AWARE AGGREGATION

**Uncertainty-Aware Weighted Aggregation (UA-WA).** We first introduce the uncertainty-aware aggregation idea in a simplified setup where the reasoning steps $s^i_{\{1:T\},k,m}$ are omitted and the task involves only questions $q$ and answers $a$. At round $k$, each of the $L$ clients uploads a predicted answer $a^i_{k+1,m}$ together with an associated uncertainty score $u^i_{k+1,m} \geq 0$ for query $q_m$. The server then calculates a weight for each client prediction using a temperature-scaled softmax over the negative uncertainties, so that predictions with lower uncertainty receive greater weight:

$$w^i_{k+1,m} = \frac{\exp(-u^i_{k+1,m}/\tau)}{\sum^L_{j=1} \exp(-u^j_{k+1,m}/\tau)}, \quad \tau > 0, \tag{3.3}$$

where $\tau$ is the temperature parameter. For each candidate answer $a \in \mathcal{A}$, the aggregated score is defined as $S_{k+1,m}(a) = \sum^L_{i=1} w^i_{k+1,m} \cdot \mathbf{1}\{a^i_{k+1,m} = a\}$, and the final prediction is selected as $a_{k+1,m} = \arg\max_{a \in \mathcal{A}} S_{k+1,m}(a)$. The updated query–answer set is then $Q_{k+1} = \{(q_m, a_{k+1,m})\}^M_{m=1}$.

**Remark 3.4.** The vanilla aggregation scheme is recovered as a special case of Eq. (3.3) by setting $w^i_{k+1,m} = 1$ for all $i$, which reduces the procedure to simple majority voting. The uncertainty-aware variant generalizes this by weighting each client's answer according to its estimated uncertainty. This is particularly useful when an infrequent answer is nonetheless produced with consistently low uncertainty. In such cases, Eq. (3.3) naturally upweights reliable yet rare signals while downweighting responses with high uncertainty. This highlights the role of uncertainty as an effective calibration mechanism within the aggregation process.

**Uncertainty-Aware Self-Critique Aggregation (UA-SCA).** We now consider the general setting where the reasoning steps $s^i_{\{1:T\},k,m}$ are explicitly generated. To address the challenge of aggregating reasoning paths—which may either align or conflict—we propose the *UA-SCA* method, summarized in Algorithm 2.

- Given client submissions $\{(s^i_{\{1:T\},k,m}, a^i_{k,m}, u^i_{k,m})\}^L_{i=1}$, UA-SCA first groups responses according to their final answers $a_m$, thereby separating consistent outputs from conflicting ones.

- Within each group, the server LLM produces a representative summary $S_G$ that captures the dominant reasoning pattern, denoted by a $\mathsf{Summarize}$ module.

- Each reasoning–answer pair $(s^i_{\{1:T\},k,m}, a^i_{k,m})$ is then revised using summaries from other groups, allowing cross-group insights to resolve inconsistencies and enrich reasoning, denoted by a $\mathsf{SelfCritique}$ module.

- Finally, the revised pairs $(\widehat{s}^i_{\{1:T\},k,m}, \widehat{a}^i_{k,m})$ and weights $w_i$ are taken as input to be aggregated by the server LLM, denoted by an $\mathsf{Aggregate}$ module. Here $w_i$ are computed from the uncertainty scores $u_m$ as in (3.3), following the UA-WA step we have studied in the simplified setting.

**Remark 3.5.** We adopt token-level entropy as the uncertainty measure because it is derived directly from the model's native token probabilities and requires no additional parameters, calibration steps, or auxiliary models. This design keeps the uncertainty signal comparable across heterogeneous client models and aligns with the privacy and computational constraints of the federated setting. Further details on the computation of uncertainty scores are provided in Section E.5.

**Remark 3.6.** FERA does not inherently require strict synchronous participation. In asynchronous settings, Algorithm 1 can be adapted by modifying line 8 to aggregate responses only from the subset of clients whose updates arrive within the current round. While a fully asynchronous variant would require a redesigned aggregation rule that jointly accounts for both uncertainty and update staleness, such an extension is orthogonal to the main focus of this work. Our goal is to introduce a training-free federated reasoning framework that leverages client-side data characteristics to guide server decision making.

**Remark 3.7.** The $\mathsf{SelfCritique}$ module is applied to assess whether a given reasoning path is meaningful by leveraging other answers as references. This idea is related to the *multi-agent debate* framework studied by Du et al. (2023). In our UA-SCA, the output of $\mathsf{SelfCritique}$ serves as an intermediate step that guides the subsequent aggregation of reasoning paths.

### 3.2 Theoretical Analysis for Uncertainty-Aware Aggregation

To further demonstrate the effectiveness of the uncertainty-aware aggregation framework, we provide a theoretical analysis of Algorithm 1 under the simplified setting, where the LLM predicts only the final answer without generating intermediate reasoning steps. We focus on its performance for linear regression tasks, where the question $q$ is a $d$-dimensional vector and the answer $a$ is a real number.

**LLM Setup.** Following Zhang et al. (2023), we assume all the clients use an identical LLM. We focus on a single-layer linear self-attention (LSA) model, which is more amenable to theoretical analysis while still capable of in-context learning linear models Zhang et al. (2023); Ahn et al. (2023); Li et al. (2023). Due to the space limit, we defer the details of definition and pretraining of LSA to Appendix C. At each client, the LLM processes a sequence of in-context examples $\{(q_i, a_i)\}^T_{i=1}$ together with a query covariate $q_{\mathsf{query}}$, and produces a prediction $a_{\mathsf{query}}$.

**Initialization & Aggregation Setup.** We assume that each $q_m$ and $q^i_n \sim N(0, \Lambda)$, where $\Lambda \succ 0 \in \mathbb{R}^{d \times d}$, the same as the pretraining distribution. To show the heterogeneity of uncertainty, for $i$-th client, we assume its answer $a^i_n = (q^i_n)^\top \theta + \epsilon^i_n, \epsilon^i_n \sim N(0, \sigma^2_i), \|\theta\| \leq 1$. Here $\sigma^2_i$ are the variances of the answers at $i$-th client, which are different. For the aggregation step, we employ weighted average aggregation: $a_{k+1,m} = \sum^L_{i=1} w^i_{k+1,m} a^i_{k+1,m}$.

**Theoretical Results.** Now we present our main theorem addressing the following two questions 1) Does our iterative update scheme impact the algorithm's performance? and 2) What is the best aggregation strategy in terms of the selection of $w^i_{k,m}$?

**Theorem 3.8.** Set the weights $w^i_{k,m} := w^i_m$ to be weights independent of the rounds and the initial answers $a_{0,m} = 0$. Then for Algorithm 1, at every round $k \in [K]$, we have:

- $a_{k,m} = \theta^\top_k q_m$ for all $m \in [M]$.

- With probability at least $1 - \delta$ for some $\delta \in (0, 1)$, the difference between $\theta_k$ and $\theta$ can be bounded by

$$\|\theta_k - \theta\| \leq O\left(\sqrt{\frac{d \log(L\delta^{-1})}{\min\{M, N\}}} + \sqrt{\frac{d \log(L\delta^{-1})}{N}} + \sqrt{\frac{d \log(L\delta^{-1})}{N}} \lambda^{-1/2}_{\min}(\Lambda) \sum^L_{i=1} w^i_m \sigma_i\right). \quad (3.4)$$

Theorem 3.8 suggests the following things. First, under the simplified linear regression task, our Algorithm 1 could converge to the ground truth linear function $q^\top \theta$ when the number of examples

$M, N$ are sufficiently large, which aligns with our expectation. Second, (3.4) suggests that the optimal weight selection should be inverse to the uncertainty $\sigma_i$, which aligns our empirical selection in (3.3) when $\tau \to \infty$, and our experiment results for FERA in Section 4.4.

## 4 EXPERIMENT

In this section, we first introduce the overall experimental setup. We then present a series of experiments designed to answer specific research questions, with each question and its corresponding results discussed in dedicated subsections.

- **RQ1.** How do FERA and its variants perform across benchmarks that target diverse reasoning tasks, including general and mathematical reasoning, compared to existing baseline methods?
- **RQ2.** What is the effect of techniques such as Uncertainty-Aware Aggregation and Iterative Refinement on the performance of FERA?
- **RQ3.** How do experimental factors such as the choice of backbone language model, number of clients, degree of data heterogeneity, and in-context length affect the performance of FERA?

### 4.1 EXPERIMENTAL SETUP

**Benchmarks.** We evaluate our approach on three established reasoning benchmarks: MMLU-Pro (Wang et al., 2024b), AQUA-RAT (Ling et al., 2017), and GSM8K (Cobbe et al., 2021). MMLU-Pro covers general reasoning across diverse domains, while AQUA-RAT and GSM8K target mathematical reasoning. Collectively, these benchmarks provide rigorous tests for evaluating the reasoning capabilities of large language models (Yang et al., 2025a; Team et al., 2025). In the following, we describe the construction of the server and client datasets in detail.

- **MMLU-Pro.** MMLU-Pro evaluates language models on over 12,000 college-level multiple-choice questions spanning 14 topics (Zeng et al., 2025). Following Du et al. (2023), we select five questions per category to form the server-side query set, with the remainder assigned to clients. To model varying degrees of data heterogeneity, client datasets are partitioned using a Dirichlet distribution (Hsu et al., 2019), where label distributions are sampled from $q \sim \mathrm{Dir}(\alpha p)$. We consider $\alpha \in \{1.0, 10, 100\}$ to represent increasing levels of uniformity.
- **AQUA-RAT.** AQUA-RAT consists of over 100,000 algebraic word problems with multiple-choice answers and natural language rationales. It emphasizes multi-step arithmetic and logical reasoning. We randomly select 70 problems for the server query set and assign the rest to clients. As the dataset lacks category labels, the data is partitioned uniformly, representing a homogeneous setting.
- **GSM8K.** GSM8K contains 8,500 grade school math problems with CoT solutions (Cobbe et al., 2021). We adopt the same protocol as for AQUA-RAT: 70 problems are used for the server query set, and the remainder are uniformly distributed among clients. The absence of category labels results in a homogeneous partitioning.

**Evaluation Metrics.** For all benchmarks, we follow standard practice and report accuracy as the primary evaluation metric (Team et al., 2023; Touvron et al., 2023b). Additionally, we compute the communication costs for each algorithm. Details of the cost calculation procedure are provided in Appendix E.3.

**Backbone Architectures.** In our main experiments, we evaluate two client-side backbone models: Qwen3-4B (Yang et al., 2025b) and LLaMA-3.1-8B (Dubey et al., 2024) by following Du et al. (2023); Huang et al. (2025). For FERA, we follow the setup of Du et al. (2023) and employ GPT-4o-mini as the server model to aggregate client responses.

### 4.2 BASELINES

**External Baselines.** We evaluate FERA against two categories of baselines: federated learning (FL) methods and parameter-free methods. The FL baselines include **FedAvg** (McMahan et al., 2017; Ye et al., 2024) and **FloRA** (Wang et al., 2024c), while the parameter-free baseline is **LLM-Debate** (Du et al., 2023). In addition, we consider a **Server-only** baseline, where the server LLM is directly applied to the benchmarks without any client collaboration.

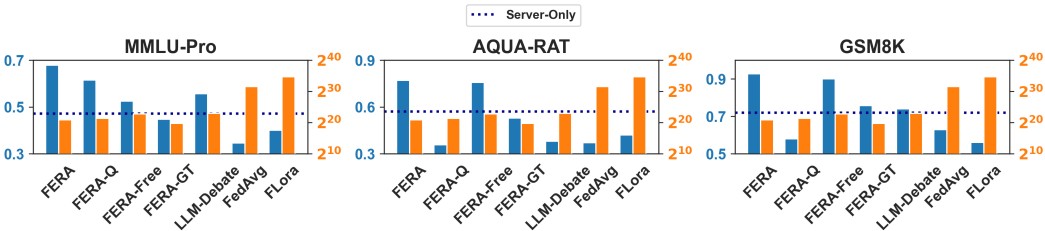

Figure 2: Performance comparison of FERA and its variants against other baselines across three benchmarks: MMLU-Pro, AQUA-RAT, and GSM8K. For MMLU-Pro, we set the Dirichlet concentration parameter to $\alpha = 10.0$ to simulate moderate client-level data heterogeneity.

**FERA Variants.** To construct a comprehensive evaluation framework for FERA, we introduce several additional baseline variants, each designed to isolate and analyze different components of the system. Detailed implementations are provided in Appendix D.5:

- **FERA-GT**: This variant uses client-provided in-context examples with *ground-truth answers* that remain fixed throughout the process. By eliminating iterative updates, it serves as an idealized upper bound for performance.
- **FERA-Q**: This variant assesses the contribution of reasoning steps in client demonstrations. Clients receive only the *final answers*, without intermediate reasoning. During inference, the model is also expected to produce only the final answer, thereby isolating the effect of reasoning on model performance.
- **FERA-Free**: Designed for low-resource settings, this variant assumes that clients possess only question datasets without corresponding answers. It evaluates FERA's capability to operate under minimal supervision, relying solely on server-side reasoning.

### 4.3 EVALUATION RESULTS

The main results are presented in Figure 2, based on experiments where all clients use the LLaMA3.1-8B model. Due to space constraints, we report only the results for this client model in the main paper; additional results using Qwen3-4B as the client model are provided in Appendix F.1. FERA consistently achieves the best performance compared to its variants and other baseline methods. In addition to accuracy, we also report the communication cost between clients and the server. The results show that FERA maintains significantly lower communication overhead than traditional federated learning baselines, demonstrating its efficiency in both performance and communication.

**Impact of Iterative Refinement.** Figure 2 shows that FERA consistently improves with an increasing number of interaction rounds, outperforming both the Server-Only baseline and FERA-GT. Unlike FERA, which refines responses iteratively through server-client communication, FERA-GT completes reasoning in a single round using a fixed local context. The superior performance of FERA highlights the benefit of iterative refinement: with each round, the model better integrates distributed knowledge and updates context representations. For results under additional settings with varying numbers of rounds, see Appendix F.1.

**Impact of Reasoning Process.** FERA-Q restricts the LLM to predicting only final answers without intermediate reasoning steps. This constraint causes noticeable performance drops, particularly on mathematical benchmarks like MMLU-Pro and AQUA-RAT, where multi-step reasoning is essential. By excluding intermediate steps, the model loses the structured reasoning path needed for complex problem solving, reducing both accuracy and interpretability. These results underscore the importance of step-by-step reasoning in logic-intensive domains.

### 4.4 ABLATION STUDIES

We conduct ablation studies to assess the contribution of key components. Due to space constraints, we focus on Uncertainty-Aware Aggregation, Demonstration Selection Strategy, Iterative Refine-

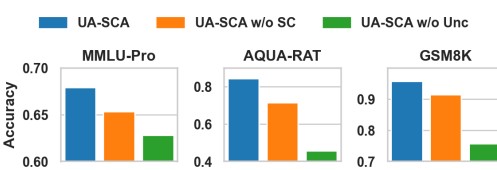

Figure 3: Performance of FERA under different aggregation strategies across different benchmarks.

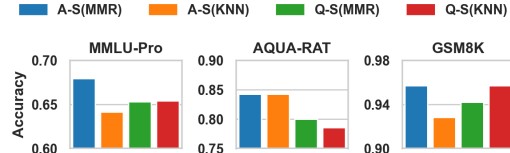

Figure 4: Performance of FERA under different demonstration selection strategies across MMLU-Pro, AQUA-RAT, and GSM8K.

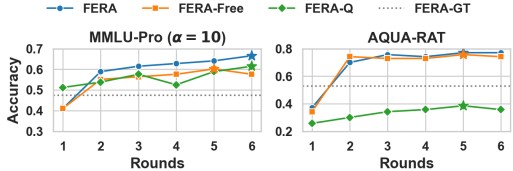

Figure 5: Effect of interaction round count on FERA and FERA-Free in the MMLU-Pro benchmark.

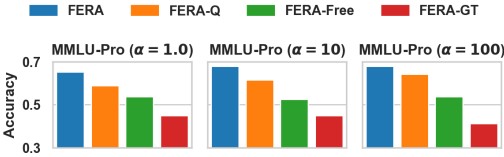

Figure 6: Impact of client-level heterogeneity on FERA and its variants in MMLU-Pro benchmark.

ment and Data Heterogeneity. Additional ablations—Client Model Capacity, Local Data Quality, Demonstration Quantity, and the Number of Clients—are provided in Appendix F.2.

**Effect of Uncertainty-Aware Aggregation.** We compare three server-side aggregation strategies in FERA, as summarized in Figure 3: (i) UA-SCA; (ii) UA-SCA without Self-Critique, where client submissions are directly aggregated without revision; and (iii) UA-SCA without Uncertainty, where the final aggregation does not incorporate the uncertainty weights $w_i$. The results show that the full UA-SCA consistently outperforms both ablations across reasoning benchmarks, highlighting the complementary benefits of uncertainty weighting and self-critique. Notably, even without self-critique, incorporating uncertainty yields clear improvements over naive aggregation, which aligns our theoretical findings in Theorem 3.8.

**Effect of Demonstration Selection Strategy.** Figure 4 presents a comparison of four client-side demonstration selection strategies, which guides how to determine the set $\mathcal{S}_{k,Q}^i$ denoted in Step 2 and 3 of FERA. They are (i) *A-S (MMR)*, our default selection (ii) *A-S (KNN)* uses the same adaptive selector but instantiates it with $k$-nearest neighbors (KNN). Given a sample $(q, s_{1:T}, a)$ and a candidate dataset, the selector first embeds the sample, computes similarity scores to all candidates, and selects the top-$k$ nearest neighbors as demonstrations. Unlike *A-S (MMR)*, which explicitly trades off relevance and diversity, *A-S (KNN)* selects demonstrations based solely on similarity and does not encourage diversity among the chosen examples. (iii) *Q-S (MMR)*, which applies MMR based solely on question similarity; and (iv) *Q-S (KNN)*, which uses KNN based only on question similarity. Among these, *A-S (MMR)* achieves the highest accuracy, outperforming both its KNN-based variant and the question-only baselines. This performance gain highlights the effectiveness of combining MMR's relevance-diversity trade-off with the adaptive selector's incorporation of broader contextual signals beyond simple question similarity.

**Effect of Iterative Refinement.** We compare the performance of FERA, FERA-Free, and FERA-GT in terms of their iterative refinement progress, as illustrated in Figure 5. FERA and FERA-Free initially underperform FERA-GT, since fixed, high-quality local exemplars give the latter an early advantage. Over additional rounds, however, uncertainty-aware demonstration selection and weighted aggregation progressively reduce noise in client updates, yielding consistent improvements that narrow the gap with FERA-GT. These results validate the effectiveness of FERA's iterative refinement in handling heterogeneous data.

**Effect of Data Heterogeneity.** We evaluate cross-client heterogeneity on MMLU-Pro by partitioning the data using a Dirichlet prior and varying the concentration parameter $\alpha$ (Hsu et al., 2019). Lower $\alpha$ values induce highly skewed, non-IID client distributions, while higher $\alpha$ values produce more homogeneous data splits. With models, prompts, and training rounds held constant, Figure 6

demonstrates that accuracy consistently increases with $\alpha$. This improvement can be attributed to two key factors: (i) more consistent demonstration retrieval across clients, reducing selection noise, and (ii) fewer conflicting updates during iterative refinement.

## 5 CONCLUSION

We introduced FERA, a federated reasoning framework that enables iterative server–client collaboration through the exchange and aggregation of reasoning steps. To address heterogeneous client data, we developed an uncertainty-aware aggregation strategy. Theoretical analysis of a simplified linear self-attention model shows that FERA's aggregation converges to ground-truth answers and that uncertainty-aware weights accelerate convergence. Extensive experiments across diverse reasoning tasks, including mathematical problem solving under heterogeneous data, demonstrate that FERA outperforms strong baselines and that each component is critical to its overall effectiveness.

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

## A  THE USE OF LARGE LANGUAGE MODELS

LLMs were used solely for language refinement. All ideas, analyses, and conclusions are the original work of the authors, who take full responsibility for the final content. In Figure 1, some icons were generated using OpenAI's ChatGPT (GPT-5). These icons are included strictly for visualization purposes and have no impact on the scientific content or conclusions of the paper.

## B  ADDITIONAL RELATED WORK

**Federated Learning in LLM.**    The growing scale of LLMs raises concerns around computational demands and data privacy (Sani et al., 2024). FL offers a solution by enabling collaborative model adaptation across decentralized data sources without sharing raw data. Several works explore FL for LLM fine-tuning and instruction tuning. Fan et al. (2023) apply FL to standard generation tasks, while Wu et al. (2024a) introduce a privacy-preserving framework designed for secure, decentralized adaptation. Kuang et al. (2024) focus on instruction tuning across heterogeneous clients to improve generalization. Beyond fine-tuning, prompting-based strategies such as Fed-SP-SC and Fed-DP-CoT (Liu et al., 2023) have shown that reasoning capabilities can be improved via federated aggregation of chain-of-thought responses, leveraging self-consistency and diversity without model updates. Collectively, these efforts underscore FL's growing role in enabling scalable, privacy-conscious LLM development.

**Reasoning Large Language Models.**    Large language models have demonstrated remarkable capabilities in complex reasoning tasks through various prompting and inference strategies. Chain-of-thought (CoT) prompting revealed that LLMs possess a latent capacity for multi-step reasoning, leading to substantial improvements in arithmetic and commonsense tasks (Yoran et al., 2023; Yao et al., 2023a; Chu et al., 2023). Follow-up studies extended this idea in several ways. Zero-shot CoT requires only the addition of "Let us think step by step" yet recovers much of the original benefit (Wei et al., 2022). Self-consistency generates multiple reasoning paths and selects the most frequent answer among them (Wang et al., 2022). Least-to-most prompting decomposes difficult problems into a sequence of simpler subquestions that are solved in order (Zhou et al., 2022). ReAct interleaves internal thoughts with environment actions, thereby combining reasoning and tool use (Yao et al., 2023b). More recently, Shinn et al. (2023) enable models to iteratively critique and refine their own outputs, further enhancing reliability and accuracy.

**Federated Reasoning for Large Language Models.**    Federated reasoning with large language models (LLMs) has been explored through training-driven approaches that update parameters in distributed settings. Full-parameter methods such as FedKSeed reduce bandwidth by transmitting random seeds and scalar gradients rather than full model weights (Qin et al., 2023). Parameter-efficient methods include FLoRA, which aggregates heterogeneous low-rank adapters (Wang et al., 2024c), FedSP, which exchanges lightweight soft prompts while preserving server-side privacy (Dong et al., 2023), and FedBiOT, which splits models into emulator–adapter pairs optimized via bi-level frameworks (Wu et al., 2024a). Task-specific frameworks have also been developed, including FedIT for instruction tuning (Zhang et al., 2024a), FedID for interactive distillation (Ma et al., 2023), and federated reinforcement learning from human feedback using lightweight preference selector aggregation. These advances show that federated training can preserve privacy and communication efficiency while remaining competitive with centralized training (Wei et al., 2025; Wu et al., 2024b).

In contrast, training-free approaches aim to enhance reasoning without parameter updates. Liu et al. (2023) leverage synonymous user questions with self-consistency voting and chain-of-thought prompting, though performance depends heavily on data homogeneity and retrieval quality. Chen et al. (2025) aggregate textual feedback into shared prompts using a density-based method, but struggle with misaligned prompts under heterogeneous data. Other in-context learning approaches (Wang et al., 2025a; Wu et al., 2024c) remain limited to simple QA and tool-use tasks, while debate-style frameworks (Du et al., 2023) risk privacy leakage through direct client communication and underutilize local datasets. Together, these lines of work highlight a trade-off: training-driven methods achieve stronger performance at higher communication and optimization costs, whereas training-free methods are lightweight but less robust under heterogeneous data and complex reasoning, motivating the development of new frameworks such as FERA.

**Uncertainty used in Federated Learning.** Recent work has addressed key challenges in FL through uncertainty-based approaches. To handle data heterogeneity, Koutsoubis et al. (2025) survey privacy-preserving methods that integrate uncertainty quantification in federated medical imaging, demonstrating how uncertainty metrics guide robust aggregation under non-IID conditions. Addressing the complementary challenge of model transparency, Zhang & Yu (2025) propose UncertainXFL, which incorporates uncertainty-aware explanations directly into the aggregation process to enhance interpretability. Model heterogeneity presents another significant challenge that has been tackled through uncertainty-guided knowledge transfer mechanisms. Wang et al. (2024a) develop FedType, which leverages proxy models with uncertainty-based distillation to bridge architectural differences, while Zhang et al. (2024b) introduce UEFL, a dynamic approach that adapts discrete codebooks based on uncertainty measures to accommodate diverse data distributions across silos.

**Uncertainty used in Large Language Models.** LLMs have focused on both quantifying and leveraging uncertainty for enhanced reliability. Hou et al. (2023) pioneer the decomposition of aleatoric and epistemic uncertainty through ensemble methods over clarified input variants, achieving interpretable uncertainty estimates without requiring architectural modifications. This decomposition framework provides a foundation for more nuanced uncertainty-aware decision-making in downstream tasks. Building on this quantification approach, Huang et al. (2024) demonstrate that output inconsistencies under label injection scenarios effectively capture intrinsic model uncertainty—a finding that directly enables active learning strategies for optimal in-context example selection. Further investigating uncertainty dynamics, Wang et al. (2025b) reveal that increased exposure to in-context examples systematically reduces predictive uncertainty, with particularly pronounced effects on epistemic uncertainty. This reduction mechanism explains the observed improvements in both model confidence and accuracy as context size increases. Integrating these theoretical insights into practice, Yang et al. (2023) operationalize an uncertainty-aware framework that empowers models with adaptive behavior: either self-correcting predictions when uncertainty is manageable or abstaining entirely when uncertainty exceeds predefined reliability thresholds.

## C  PROOF IN SECTION 3

### C.1  PRELIMINARIES FOR THE THEORETICAL ANALYSIS

In this section, we lay out the basic formulation of in-context learning (ICL) for function classes, building on Garg et al. (2022); Zhang et al. (2023).

In ICL, the model processes and exploits sequential input called a prompt. A prompt consists of a sequence of input–output pairs $(x_1, y_1, \ldots, x_N, y_N, x_{\text{query}})$, where each $y_i$ corresponds to the evaluation of an unknown target function $h$ at $x_i$. Given such a sequence, the model's task is to infer useful information from the examples and generate a prediction $\widehat{y}(x_{\text{query}})$ for the query point $x_{\text{query}}$, aiming for $\widehat{y}(x_{\text{query}}) \approx h(x_{\text{query}})$.

We now present a formal definition for models that learn from in-context examples, following Zhang et al. (2023).

**Definition C.1** ((Definition 3.1 in Zhang et al. (2023)) Trained on in-context examples). Let $\mathcal{D}_x$ be a distribution over an input space $\mathcal{X}$, $\mathcal{H} \subset \mathcal{Y}^{\mathcal{X}}$ a set of functions $\mathcal{X} \to \mathcal{Y}$, and $\mathcal{D}_{\mathcal{H}}$ a distribution over functions in $\mathcal{H}$. Let $\ell : \mathcal{Y} \times \mathcal{Y} \to \mathbb{R}$ be a loss function. Let $\mathcal{S} = \cup_{n \in \mathbb{N}} \{(x_1, y_1, \ldots, x_n, y_n) : x_i \in \mathcal{X}, y_i \in \mathcal{Y}\}$ be the set of finite-length sequences of $(x, y)$ pairs and let

$$\mathcal{F}_{\Theta} = \{f_{\theta} : \mathcal{S} \times \mathcal{X} \to \mathcal{Y}, \theta \in \Theta\}$$

be a class of functions parameterized by $\theta$ in some set $\Theta$. For $T > 0$, we say that a model $f : \mathcal{S} \times \mathcal{X} \to \mathcal{Y}$ is *trained on in-context examples of functions in $\mathcal{H}$ under loss $\ell$ w.r.t. $(\mathcal{D}_{\mathcal{H}}, \mathcal{D}_x)$* if $f = f_{\theta^*}$ where $\theta^* \in \Theta$ satisfies

$$\theta^* \in \operatorname{argmin}_{\theta \in \Theta} \mathbb{E}_{P=(x_1, h(x_1), \ldots, x_T, h(x_T), x_{\text{query}})} \left[\ell\left(f_{\theta}(P), h(x_{\text{query}})\right)\right], \quad (\text{C.1})$$

where $x_i, x_{\text{query}} \overset{\text{i.i.d.}}{\sim} \mathcal{D}_x$ and $h \sim \mathcal{D}_{\mathcal{H}}$ are independent. We call $T$ the *length of the prompts seen during training.*

Let $E \in \mathbb{R}^{d_e \times d_T}$ denote an embedding matrix associated with a prompt $P = (x_1, y_1, \ldots, x_T, y_T, x_{\text{query}})$. Following Zhang et al. (2023), we form each column from $(x_i, y_i)^{\top} \in$

$\mathbb{R}^{d+1}$ for $i = 1, \ldots, T$, and use $(x_{\mathsf{query}}, 0)^\top$ as the final column. For $x_i \in \mathbb{R}^d$ and $y_i \in \mathbb{R}$, we have $d_e = d + 1$ and $d_T = T + 1$. Under the prompt representation $P$, the embedding matrix can be written as

$$E = E(P) = \begin{pmatrix} x_1 & x_2 & \cdots & x_T & x_{\mathsf{query}} \\ y_1 & y_2 & \cdots & y_T & 0 \end{pmatrix} \in \mathbb{R}^{(d+1) \times (T+1)}. \tag{C.2}$$

We introduce weight matrices $W^Q, W^K \in \mathbb{R}^{d_k \times d_e}$ for the query and key, $W^V \in \mathbb{R}^{d_v \times d_e}$ for the value, $W^P \in \mathbb{R}^{d_e \times d_v}$ for projection, and a normalization constant $\rho > 0$.

For the sake of tractability in theoretical analysis, following Zhang et al. (2023), we consider a single-layer *linear self-attention (LSA)* model. This variant streamlines the standard self-attention by eliminating the softmax step and merging the projection matrices. Specifically, we define merged operators $W^{KQ} \in \mathbb{R}^{d_e \times d_e}$ (query–key) and $W^{PV} \in \mathbb{R}^{d_e \times d_e}$ (projection–value). Letting $\theta = (W^{KQ}, W^{PV})$, the LSA model can be formulated as

$$f_{\mathrm{LSA}}(E; \theta) = E + W^{PV} E \cdot \frac{E^\top W^{KQ} E}{\rho}. \tag{C.3}$$

The prediction corresponding to the query token $x_{\mathsf{query}}$ is given by the bottom-right entry of $f_{\mathrm{LSA}}$:

$$\widehat{y}_{\mathsf{query}} = \widehat{y}_{\mathsf{query}}(E; \theta) = [f_{\mathrm{LSA}}(E; \theta)]_{(d+1),(T+1)}.$$

To streamline the theoretical development, we restrict attention to in-context learning with linear predictors, in line with the setup of Zhang et al. (2023). We assume that all clients share the same sampling procedure for generating training prompts. Let $\Lambda$ be a positive definite covariance matrix. For each task indexed by $\tau \in \mathbb{N}$, we construct a training prompt

$$P_\tau = (x_{\tau,1}, h_\tau(x_{\tau,1}), \ldots, x_{\tau,T}, h_\tau(x_{\tau,T}), x_{\tau,\mathsf{query}}).$$

The task-specific parameter $w_\tau$ is drawn independently from $\mathcal{N}(0, I_d)$, the inputs $x_{\tau,i}$ and $x_{\tau,\mathsf{query}}$ are sampled i.i.d. from $\mathcal{N}(0, \Lambda)$, and the labels are defined by the linear rule $h_\tau(x) = \langle w_\tau, x \rangle$.

Each prompt $P_\tau$ is then converted into an embedding matrix $E_\tau$ using the transformation in Eq. (C.2):

$$E_\tau = \begin{pmatrix} x_{\tau,1} & x_{\tau,2} & \cdots & x_{\tau,T} & x_{\tau,\mathsf{query}} \\ \langle w_\tau, x_{\tau,1} \rangle & \langle w_\tau, x_{\tau,2} \rangle & \cdots & \langle w_\tau, x_{\tau,T} \rangle & 0 \end{pmatrix} \in \mathbb{R}^{(d+1) \times (T+1)}.$$

The empirical risk evaluated over $B$ independent prompts is given by

$$\widehat{L}(\theta) = \frac{1}{2B} \sum_{\tau=1}^{B} \left( \widehat{y}_{\tau,\mathsf{query}} - \langle w_\tau, x_{\tau,\mathsf{query}} \rangle \right)^2. \tag{C.4}$$

To study the limiting behavior, we introduce the population loss obtained as $B \to \infty$:

$$L(\theta) = \lim_{B \to \infty} \widehat{L}(\theta) = \frac{1}{2} \mathbb{E}_{w_\tau, x_{\tau,1}, \ldots, x_{\tau,T}, x_{\tau,\mathsf{query}}} \left[ \left( \widehat{y}_{\tau,\mathsf{query}} - \langle w_\tau, x_{\tau,\mathsf{query}} \rangle \right)^2 \right]. \tag{C.5}$$

Here the expectation is taken over the random task weight $w_\tau \sim \mathcal{N}(0, I_d)$ and the covariates $\{x_{\tau,i}\}_{i=1}^T \cup \{x_{\tau,\mathsf{query}}\}$, drawn i.i.d. from $\mathcal{N}(0, \Lambda)$.

We analyze optimization via the *gradient flow* framework, which describes the continuous-time limit of gradient descent with infinitesimal step size. The dynamics of the parameters follow the ODE

$$\frac{\mathrm{d}}{\mathrm{d}t} \theta = -\nabla L(\theta). \tag{C.6}$$

In the remainder, we study gradient flow trajectories under initializations that satisfy the following assumptions in Zhang et al. (2023):

**Assumption C.2** (Initialization (Zhang et al. (2023))). Let $\sigma > 0$ be a parameter, and let $\Theta \in \mathbb{R}^{d \times d}$ be any matrix satisfying $\|\Theta\Theta^\top\|_F = 1$ and $\Theta\Lambda \neq 0_{d \times d}$. We assume

$$W^{PV}(0) = \sigma \begin{pmatrix} 0_{d \times d} & 0_d \\ 0_d^\top & 1 \end{pmatrix}, \quad W^{KQ}(0) = \sigma \begin{pmatrix} \Theta\Theta^\top & 0_d \\ 0_d^\top & 0 \end{pmatrix}. \tag{C.7}$$

under suitable initialization, gradient flow will converge to a global optimum.

**Theorem C.3** ((Theorem 4.1 of Zhang et al. (2023)) Convergence and limits)**.** Consider the gradient flow of the linear self-attention network $f_{\mathsf{LSA}}$ defined in (C.3) over the population loss (C.5). Suppose the initialization satisfies Assumption C.2 with initialization scale $\sigma > 0$ satisfying $\sigma^2 \|\Gamma\|_{op} \sqrt{d} < 2$ where we have defined

$$\Gamma := \left(1 + \frac{1}{T}\right)\Lambda + \frac{1}{T}\operatorname{tr}(\Lambda)I_d \in \mathbb{R}^{d \times d}.$$

Then, the gradient flow converges to a global minimum of the population loss (C.5). Moreover, $W^{PV}$ and $W^{KQ}$ converge to $W_*^{PV}$ and $W_*^{KQ}$ respectively, where

$$W_*^{KQ} = \left[\operatorname{tr}\left(\Gamma^{-2}\right)\right]^{-\frac{1}{4}} \begin{pmatrix} \Gamma^{-1} & 0_d \\ 0_d^\top & 0 \end{pmatrix}, \qquad W_*^{PV} = \left[\operatorname{tr}\left(\Gamma^{-2}\right)\right]^{\frac{1}{4}} \begin{pmatrix} 0_{d \times d} & 0_d \\ 0_d^\top & 1 \end{pmatrix}. \tag{C.8}$$

At the global optimum with parameters $W_*^{KQ}$ and $W_*^{PV}$, the prediction for the query token takes the form

$$\widehat{y}_{\mathsf{query}} = \begin{pmatrix} 0_d^\top & 1 \end{pmatrix} \begin{pmatrix} \frac{1}{M}\sum_{i=1}^M x_i x_i^\top + \frac{1}{M}x_{\mathsf{query}}x_{\mathsf{query}}^\top & \frac{1}{M}\sum_{i=1}^M x_i y_i \\ \frac{1}{M}\sum_{i=1}^M x_i^\top y_i & \frac{1}{M}\sum_{i=1}^M y_i^2 \end{pmatrix} \begin{pmatrix} \Gamma^{-1} & 0_d \\ 0_d^\top & 0 \end{pmatrix} \begin{pmatrix} x_{\mathsf{query}} \\ 0 \end{pmatrix}$$

$$= x_{\mathsf{query}}^\top \Gamma^{-1} \left(\frac{1}{M}\sum_{i=1}^M y_i x_i\right). \tag{C.9}$$

## C.2 Proof of Theorem 3.8

According to the above preliminary theoretical results adopted from Zhang et al. (2023), we have the following analysis.

*Proof.* First by our assumption of the client and server data, we have

$$q_n^i \sim N(0, \Lambda), \ q_m \sim N(0, \Lambda),$$

and

$$a_n^i = (q_n^i)^\top \theta + \epsilon_n^i, \epsilon_n^i \sim N(0, \sigma_i^2), \|\theta\| \le 1,$$

where the variance $\sigma_i^2$ are different from client to client, representing the heterogeneity over clients. Then we have the following inequality holds with probability at least $1 - \delta$:

$$\left\| I - \Gamma^{-1}\frac{\sum_{m=1}^M q_m(q_m)^\top}{M} \right\|_{op}$$

$$\le \left\| \Gamma^{-1}\Lambda \left( I - \Lambda^{-1}\frac{\sum_{m=1}^M q_m(q_m)^\top}{M} \right) \right\|_{op} + \left\| I - \Gamma^{-1}\Lambda \right\|_{op}$$

$$\le \|\Gamma^{-1}\Lambda\| \cdot 4 \cdot \left( \sqrt{\frac{d\log\delta^{-1}}{M}} + \frac{d\log\delta^{-1}}{M} \right) + \left\| I - \Gamma^{-1}\Lambda \right\|_{op}$$

$$\le 10 \cdot \sqrt{\frac{d\log\delta^{-1}}{M}}, \tag{C.10}$$

where for the second inequality, we use the standard matrix concentration inequality, for the last one, we use the fact that $T$ is large enough and $M \ge 4d\log\delta^{-1}$, where

$$\left\| I - \Gamma^{-1}\Lambda \right\|_{op} = \frac{\lambda_{\min}(\Lambda) + \operatorname{tr}(\Lambda)}{T\lambda_{\min}(\Lambda) + \operatorname{tr}(\Lambda)} \le \sqrt{\frac{d\log\delta^{-1}}{M}} \le \frac{1}{4}.$$

Similarly, with probability at least $1 - \delta$, we have for all $i \in [L]$,

$$\left\| I - \Gamma^{-1}\frac{\sum_{n=1}^N q_n^i(q_n^i)^\top}{N} \right\|_{op} \le 10 \cdot \sqrt{\frac{d\log(L\delta^{-1})}{N}}. \tag{C.11}$$

First, with Eq.(C.9) and the algorithm design of Algorithm 1, we have the following guarantees

$$\widehat{a}_{k,n}^i = (q_n^i)^\top \Gamma^{-1} \left( \frac{1}{M} \sum_{m=1}^M q_m a_{k,m} \right), \tag{C.12}$$

$$a_{k+1,m}^i = q_m^\top \Gamma^{-1} \left( \frac{1}{2N} \sum_{n=1}^N q_n^i (a_n^i + \widehat{a}_{k,n}^i) \right). \tag{C.13}$$

Then we can prove the theorem by induction.

Assume $a_{k,m} = \theta_k^\top q_m$ holds for episode $k$, which trivially holds at episode 1 with $\theta_1 = 0$ as $a_{1,m} = 0, \forall m \in [M]$. According to Eq.(C.12), Eq.(C.13), and $a_{k+1,m} = \sum_{i=1}^L w_{k+1,m}^i a_{k+1,m}^i$, we have

$$\begin{aligned}
a_{k+1,m}^i &= q_m^\top \Gamma^{-1} \left( \frac{1}{2N} \sum_{n=1}^N q_n^i (a_n^i + \widehat{a}_{k,n}^i) \right) \\
&= q_m^\top \Gamma^{-1} \left( \frac{1}{2N} \left( \sum_{n=1}^N q_n^i a_n^i + \sum_{n=1}^N q_n^i (q_n^i)^\top \Gamma^{-1} \left( \frac{1}{M} \sum_{m=1}^M q_m a_{k,m} \right) \right) \right) \\
&= q_m^\top \Gamma^{-1} \left( \frac{1}{2N} \left( \sum_{n=1}^N q_n^i a_n^i + \sum_{n=1}^N q_n^i (q_n^i)^\top \Gamma^{-1} \left( \frac{1}{M} \sum_{m=1}^M q_m q_m^\top \right) \theta_k \right) \right),
\end{aligned}$$

Then, we have

$$\begin{aligned}
a_{k+1,m} &= \sum_{i=1}^L w_{k+1,m}^i a_{k+1,m}^i \\
&= \sum_{i=1}^L w_{k+1,m}^i q_m^\top \Gamma^{-1} \left( \frac{1}{2N} \left( \sum_{n=1}^N q_n^i a_n^i + \sum_{n=1}^N q_n^i (q_n^i)^\top \Gamma^{-1} \left( \frac{1}{M} \sum_{m=1}^M q_m q_m^\top \right) \theta_k \right) \right) \\
&= q_m^\top \left( \Gamma^{-1} \frac{\sum_{i=1}^L w_{k+1,m}^i \sum_{n=1}^N q_n^i a_n^i}{2N} + \frac{\sum_{i=1}^L w_{k+1,m}^i \Gamma^{-1} \sum_{n=1}^N q_n^i (q_n^i)^\top \Gamma^{-1} \sum_{m=1}^M q_m q_m^\top}{2NM} \cdot \theta_k \right) \\
&= q_m^\top \left( \underbrace{\frac{1}{2} \sum_{i=1}^L w_{k+1,m}^i \left( \Gamma^{-1} \frac{\sum_{n=1}^N q_n^i (q_n^i)^\top}{N} \right) \left( \Gamma^{-1} \frac{\sum_{m=1}^M q_m (q_m)^\top}{M} \right)}_{H_k} \cdot \theta_k \right. \\
&\quad \left. + \underbrace{\frac{1}{2} \cdot \sum_{i=1}^L w_{k+1,m}^i \Gamma^{-1} \frac{\sum_{n=1}^N q_n^i a_n^i}{N}}_{\bar{\theta}_k} \right). \tag{C.14}
\end{aligned}$$

Then appreately, we have shown that for each $k$, $a_{k,m} = \theta_k^\top q_m$ can be written as a linear function of the query $q_m$. Next we bound $\bar{\theta}_k$ and $H_k$ separately. We have

$$\begin{aligned}
\bar{\theta}_k &= \sum_{i=1}^L w_{k+1,m}^i \Gamma^{-1} \frac{\sum_{n=1}^N q_n^i (q_n^i)^\top \theta + q_n^i \epsilon_n^i}{N} \\
&= \theta - \underbrace{\sum_{i=1}^L w_{k+1,m}^i \left( I - \Gamma^{-1} \frac{\sum_{n=1}^N q_n^i (q_n^i)^\top}{N} \right) \theta}_{A_{k,m}} + \underbrace{\sum_{i=1}^L w_{k+1,m}^i \Gamma^{-1} \frac{\sum_{n=1}^N q_n^i \epsilon_n^i}{N}}_{B_{k,m}}, \tag{C.15}
\end{aligned}$$

and

$$
H_k = \sum_{i=1}^{L} w_{k+1,m}^i \left( I - \left( I - \Gamma^{-1} \frac{\sum_{n=1}^{N} q_n^i (q_n^i)^\top}{N} \right) \right) \left( I - \left( I - \Gamma^{-1} \frac{\sum_{m=1}^{M} q_m (q_m)^\top}{M} \right) \right)
$$

$$
= I - \underbrace{\sum_{i=1}^{L} w_{k+1,m}^i \left( I - \Gamma^{-1} \frac{\sum_{n=1}^{N} q_n^i (q_n^i)^\top}{N} \right) - \sum_{i=1}^{L} w_{k+1,m}^i \left( I - \Gamma^{-1} \frac{\sum_{m=1}^{M} q_m (q_m)^\top}{M} \right)}_{C_{k,m}}
$$

$$
+ \underbrace{\sum_{i=1}^{L} w_{k+1,m}^i \left( I - \Gamma^{-1} \frac{\sum_{n=1}^{N} q_n^i (q_n^i)^\top}{N} \right) \left( I - \Gamma^{-1} \frac{\sum_{m=1}^{M} q_m (q_m)^\top}{M} \right)}_{D_{k,m}}. \tag{C.16}
$$

**Bound $A_{k,m}$.** For $A_{k,m}$ by (C.11), we have

$$
\|A_{k,m}\|_{\mathrm{op}} \leq \|\theta\| \left\| I - \Gamma^{-1} \frac{\sum_{n=1}^{N} q_n^i (q_n^i)^\top}{N} \right\|_{\mathrm{op}} \leq 10 \cdot \sqrt{\frac{d \log(L\delta^{-1})}{N}}. \tag{C.17}
$$

**Bound $B_{k,m}$.** For $B_{k,m}$, recall that $q_n^i \sim N(0, \Lambda)$ and $\epsilon_n^i \sim N(0, \sigma_i^2)$, then $\Lambda^{-1/2} q_n^i \sim N(0, I)$. Then applying concentration inequality on sub-exponential random vectors, with probability at least $1 - \delta$, we have

$$
B_{k,m} := \Gamma^{-1} \Lambda^{1/2} \sum_{i=1}^{L} w_{k+1,m}^i \frac{\sum_{n=1}^{N} \Lambda^{-1/2} q_n^i \epsilon_n^i}{N}
$$

$$
\leq \|\Gamma^{-1} \Lambda^{1/2}\| \sum_{i=1}^{L} w_m^i \cdot \left\| \frac{\sum_{n=1}^{N} \Lambda^{-1/2} q_n^i \epsilon_n^i}{N} \right\|
$$

$$
\leq \lambda_{\min}^{-1/2}(\Lambda) \cdot \sum_{i=1}^{L} w_m^i \cdot 10\sigma_i \left( \sqrt{\frac{d \log(L\delta^{-1})}{N}} + \frac{d^{1.5} \log(L\delta^{-1})}{N} \right). \tag{C.18}
$$

**Bound $C_{k,m}$.** For $C_{k,m}$, using (C.10) and (C.11), we have

$$
\|C_{k,m}\|_{\mathrm{op}} \leq \left( \left\| I - \Gamma^{-1} \frac{\sum_{n=1}^{N} q_n^i (q_n^i)^\top}{N} \right\|_{\mathrm{op}} + \left\| I - \Gamma^{-1} \frac{\sum_{m=1}^{M} q_m (q_m)^\top}{M} \right\|_{\mathrm{op}} \right) \leq 20 \cdot \sqrt{\frac{d \log(L\delta^{-1})}{\min\{M, N\}}}. \tag{C.19}
$$

**Bound $D_{k,m}$.** For $D_{k,m}$, using (C.10) and (C.11), we have

$$
\|D_{k,m}\|_{\mathrm{op}} \leq \left\| I - \Gamma^{-1} \frac{\sum_{n=1}^{N} q_n^i (q_n^i)^\top}{N} \right\|_{\mathrm{op}} \cdot \left\| I - \Gamma^{-1} \frac{\sum_{m=1}^{M} q_m (q_m)^\top}{M} \right\|_{\mathrm{op}} \leq 100 \cdot \frac{d \log(L\delta^{-1})}{\min\{M, N\}}. \tag{C.20}
$$

Therefore, combining (C.17) to (C.20), we have

$$
\|\theta_{k+1} - \theta\| = \left\| \left( \frac{1}{2} + \frac{C_{k,m} - D_{k,m}}{2} \right) (\theta_k - \theta) - \frac{A_{k,m} - B_{k,m}}{2} - \frac{C_{k,m} - D_{k,m}}{2} \theta \right\|
$$

$$
\leq \left\| \frac{1}{2} + \frac{C_{k,m} - D_{k,m}}{2} \right\| \|\theta_k - \theta\| + \left\| \frac{C_{k,m} - D_{k,m}}{2} \right\| + \left\| \frac{A_{k,m} - B_{k,m}}{2} \right\|. \tag{C.21}
$$

Since

$$
\left\| \frac{C_{k,m} - D_{k,m}}{2} \right\| < 10 \cdot \sqrt{\frac{d \log(L\delta^{-1})}{\min\{M, N\}}} + 50 \cdot \frac{d \log(L\delta^{-1})}{\min\{M, N\}} < \frac{1}{4} \tag{C.22}
$$

and

$$\left\| \frac{A_{k,m} - B_{k,m}}{2} \right\| \leq 5 \cdot \sqrt{\frac{d \log(L\delta^{-1})}{N}} + 5\lambda_{\min}^{-1/2}(\Lambda) \cdot \sum_{i=1}^{L} w_m^i \sigma_i \left( \sqrt{\frac{d \log(L\delta^{-1})}{N}} + \frac{d^{1.5} \log(L\delta^{-1})}{N} \right)$$

(C.23)

Then applying recursion onto (C.21), we have that for all $k$,

$$\|\theta_k - \theta\| \leq O\left( \sqrt{\frac{d \log(L\delta^{-1})}{\min\{M, N\}}} + \sqrt{\frac{d \log(L\delta^{-1})}{N}} + \sqrt{\frac{d \log(L\delta^{-1})}{N}} \lambda_{\min}^{-1/2}(\Lambda) \sum_{i=1}^{L} w_m^i \sigma_i \right).$$

$\square$

# D PROMPT

## D.1 SERVER QUERY DATASET INITIALIZE

To initialize the query dataset, the server distributes queries to multiple clients. Each client employs its local LLM to generate the responses for the assigned queries. The responses are then returned to the server, which aggregates them to form the initial server-side query dataset. The prompts used to guide this response generation are described as follows.

---

**Server Query Initialization Prompt for MMLU-Pro & AQUA-RAT Reasoning Benchmark**

You are a knowledgeable assistant. For the following multiple-choice question, briefly explain your reasoning (no more than {sentences_limit} sentences), then end with the exact sentence: The answer is (X).

**Rules:**
1. X must be the option letter only (A/B/C/D/...). Do not include the option text.
2. Do not include any content after the final sentence.
3. Keep your entire response within {token_limit} tokens.

**Question:** {query}
**Answer:** Let's think step by step. {text}

---

**Server Query Initialization Prompt for MMLU-Pro & AQUA-RAT Standard QA Benchmark**

You are taking a multiple-choice question. Read the following question carefully and select the single best answer. Do *not* explain your reasoning. Output only the final answer choice letter (A, B, C, D, ...).

**Rules:**
1. The output must be a single uppercase letter (A/B/C/D/...) with no punctuation or extra text.
2. Do not include any explanation or content after the answer.
3. Keep the response within {token_limit} tokens.

**Question:** {query}

**Answer:**

---

---

**Server Query Initialization Prompt for GSM8K Reasoning Benchmark**

```
You are a knowledgeable assistant.  For the following math
question, briefly explain your reasoning (no more than
{sentences_limit} sentences), then end with the exact sentence:  The
answer is X.
```

**Rules:**

```
1. X must be a single numeric value (e.g., 12, -3/5, 7.25); no
   units or extra text.
2. If X is a fraction, reduce it to simplest terms; if a decimal,
   use standard form without trailing zeros.
3. Do not include any content after the final sentence.
4. Keep the entire response within {token_limit} tokens.
```

**Question:** {query}

**Answer:  Let's think step by step.**

---

### D.2 CLIENT RESPONSE GENERATION FOR SERVER QUERIES

The framework of FERA is outlined in Algorithm 1. In Step 2, each client relabels its local data by constructing prompts that incorporate demonstrations selected from the server dataset. In Step 3, the client relabels the server data by constructing prompts that include demonstrations drawn from its updated local dataset. The prompts used in Step 2 and Step 3 are described below. Unless otherwise specified, the default number of demonstrations is set to 5.

---

**Client Prediction Prompt for MMLU-Pro & AQUA-RAT Reasoning Benchmark**

```
You are tasked with answering multiple-choice math questions.
Below are several example questions with their step-by-step
reasoning and final answers.  After reviewing these examples, you
will be presented with a new question to answer.
```

**Guidelines:**

```
1. Provide clear, concise, and logically coherent step-by-step
   reasoning (at most {sentences_limit} sentences).
2. End with the exact sentence:  The answer is (X).
3. X must be the option letter only (A, B, C, D, ...); do not
   include the option text.
4. Include no additional content after the final answer sentence.
5. Keep the complete response within {token_limit} tokens.
```

**Examples:**
{examples}

**Question:**
{query}

**Answer:  Let's think step by step.**

---

---

**Client Prediction Prompt for MMLU-Pro & AQUA-RAT Standard QA Benchmark**

```
You are taking a multiple-choice question.  Below are several
example questions with their final answers.  After reviewing these
examples, you will be presented with a new question to answer.
```

**Guidelines:**

1. Read the question carefully and select the single best answer.

2. Do *not* explain your reasoning.

3. Output only the final answer choice letter (A, B, C, D, ...); do not include the option text.

4. Do not include any additional content after the answer.

**Examples:**
{examples}

**Question:**
{query}

**Answer:**

---

**Client Prediction Prompt for GSM8K Reasoning Benchmark**

```
You are tasked with answering math questions in this domain.  Below
are several example questions with their step-by-step reasoning
and final answers.  After reviewing these examples, you will be
presented with a new question to answer.
```

**Guidelines:**

1. Provide clear, concise, and logically coherent step-by-step reasoning.

2. End your response with the exact sentence:  The answer is X.

3. X must be a single numeric value (e.g., 12, -3/5, 7.25); no units or extra text.

4. If X is a fraction, reduce it to simplest terms; if a decimal, use standard form without trailing zeros.

5. Include no additional content after the final answer sentence.

6. Keep the complete response within {token_limit} tokens.

**Examples:**
{examples}

**Question:**
{query}

**Answer:**  Let's think step by step.

---

## D.3   Uncertainty-Aware Demonstration Selection

In the FERA framework, Uncertainty-Aware Demonstration Selection (UADS) is employed on the client side to determine which demonstrations are included in the prompt. At each round, embeddings are computed for both the client dataset and the server query–answer pairs. The Maximal Marginal Relevance (MMR) strategy is then applied to select demonstrations that jointly optimize similarity to the query and diversity among themselves. Since the server query answer may change across rounds, the demonstration set is updated accordingly, rendering the process adaptive. Notably, in the standard QA benchmark the answer consists solely of the final choice, whereas in the reasoning benchmark the answer additionally includes the full reasoning process leading to the final solution.

### D.4 UNCERTAINTY-AWARE AGGREGATION

In the FERA framework, Uncertainty-Aware Aggregation is employed on the server side to combine responses from clients. The weights used in this aggregation are first computed according to Equation 3.3, after which uncertainty is leveraged to guide the aggregation process. For the standard QA task, we propose the UA-WA algorithm, described in Section 3.1, while for complex reasoning tasks we design the UA-SCA algorithm, detailed in Algorithm 2. The prompts utilized in UA-SCA are presented below.

---

**Summarize**

```
You are a cognitive reasoning analyst tasked with examining
{len(reasoning_list)} reasoning responses derived from the following
question.
```

**Question:**
```
{question}
```

**Reasoning Responses:**
```
{reasoning_formatted}
```

**Analysis Objective:**
```
Provide a concise analytical characterization of this reasoning
cluster.  Identify the cognitive patterns, methodological
strategies, and structural similarities across the responses.
```

**Characterization Guidelines:**
```
1. Synthesize the distinguishing features of these reasoning
   responses
2. Emphasize their reasoning methodology and structural
   organization
3. Identify common problem-solving paradigms across responses
4. Present your analysis within {token_limit} tokens
5. Capture the essential cognitive characteristics of this cluster
```

**Your Analysis:**

---

**SelfCritique for MMLU-Pro and AQUA-RAT Benchamrk**

You are improving a reasoning response by incorporating insights from conflicting reasoning approaches.

**Original Question:**
{question}

**Target Reasoning Response (to be improved):**
{target_response}

**Alternative Approaches Summary:**
{alternatives_formatted}

**Task:**
Create an enhanced version of the target reasoning response by incorporating valuable insights from the alternative approaches. Identify reasoning elements, methodologies, or perspectives from the conflicting summaries that could strengthen the original response.

**Requirements:**
1. Use the target reasoning response as your foundation
2. Extract valuable insights from the alternative approaches
3. Integrate these insights to create a more comprehensive response
4. Maintain logical consistency throughout
5. Present only the final improved reasoning
6. Conclude with: ``The answer is (X)'' where X is the option letter only (A/B/C/D/...)
7. Exclude option text after the letter
8. Limit response to {token_limit} tokens
9. Omit meta-commentary or explanatory analysis

**Improved Response:**

**SelfCritique for GSM8K Benchamrk**

You are improving a reasoning response by incorporating insights from conflicting reasoning approaches.

**Original Question:**
{question}

**Target Reasoning Response (to be improved):**
{target_response}

**Alternative Approaches Summary:**
{alternatives_formatted}

**Task:**
Create an enhanced version of the target reasoning response by incorporating valuable insights from the alternative approaches. Identify reasoning elements, methodologies, or perspectives from the conflicting summaries that could strengthen the original response.

**Requirements:**
1. Start with the target reasoning as your foundation
2. Identify useful insights from the conflicting summaries that could improve the reasoning
3. Integrate these insights to create a stronger, more comprehensive reasoning
4. Maintain logical consistency throughout
5. Present only the final improved reasoning
6. End with "The answer is X".
7. Limit response to {token_limit} tokens
8. No meta-commentary or analysis explanation

**Improved Response:**

**Aggregation for MMLU-Pro and AQUA-RAT Benchmark**

```
You are synthesizing multiple reasoning responses to create a
single, unified reasoning path.
```

**Context:**
```
You are given several reasoning responses to the same question,
each from a different client.  A final answer has been determined
by majority vote.  Each response includes a confidence score
(higher = more confident).
```

**Question:**
```
{question}
```

**Client Responses (with confidence scores):**
```
{client_entries}
```

**Task:**
```
Produce a single, concise, professional, and logically coherent
merged reasoning response that synthesizes the reasoning leading to
the final answer.
```

**Requirements:**
```
1. Synthesize the reasoning leading to the final answer
2. Give greater weight to reasoning from higher-confidence
   responses
3. Avoid unnecessary repetition or irrelevant details
4. Keep the ENTIRE response (reasoning + final answer) within
   {token_limit} tokens
5. End with:  ``The answer is (X)'' where X is the option letter
   only (A/B/C/D/...)
6. Do not include the option text after the letter
7. Maintain professional and logical coherence throughout
```

**Merged Reasoning Response:**

**Aggregation for MMLU-Pro and AQUA-RAT Benchmark**

You are synthesizing multiple reasoning responses to create a single, unified reasoning path.

**Context:**
You are given several reasoning responses to the same question, each from a different client. Each response includes a confidence score (higher = more confident).

**Question:**
{question}

**Client Responses (with confidence scores):**
{client_entries}

**Task:**
Produce a single, concise, professional, and logically coherent merged reasoning response that synthesizes the reasoning leading to the final answer.

**Requirements:**
1. Synthesize the reasoning leading to the final answer
2. Give greater weight to reasoning from higher-confidence responses
3. Avoid unnecessary repetition or irrelevant details
4. Keep the ENTIRE response (reasoning + final answer) within {token_limit} tokens
5. End with: ``The answer is (X)'' where X is the option letter only (A/B/C/D/...)
6. Do not include the option text after the letter
7. Maintain professional and logical coherence throughout

**Merged Reasoning Response:**

---

**Aggregation for GSM8K Benchmark**

```
You are synthesizing multiple reasoning responses to create a
single, unified solution path.

Context:
You are given several reasoning responses to the same question,
each from a different client. Each response includes a confidence
score (higher = more confident).

Question:
{question}

Client Responses (with confidence scores):
{client_entries}

Task:
Produce a single, concise, professional, and logically coherent
merged reasoning response that synthesizes the reasoning leading to
the final answer.

Requirements:
1. Synthesize the reasoning leading to the final answer
2. Give greater weight to reasoning from higher-confidence
   responses
3. Avoid unnecessary repetition or irrelevant details
4. Keep the ENTIRE response (reasoning + final answer) within
   {token_limit} tokens
5. End with the exact sentence:  ``The answer is X.''
6. X must be a single numeric value (e.g., 12, -3/5, 7.25)
7. No units or extra text after the answer

Merged Reasoning Response:
```

---

## D.5 FERA VARIANTS

**FERA-GT.** To evaluate the effectiveness of FERA, we compare it with a simplified baseline, FERA-GT. In this baseline, the server issues a query to clients, who independently retrieve the top-$C$ question–answer pairs using the MMR demonstration strategy. These retrieved pairs serve as fixed context for generating client responses, which are then aggregated by the server to produce the final answer. Importantly, FERA-GT completes this process in a single communication round without iterative context refinement. In contrast, FERA employs multiple rounds of interaction, enabling dynamic context updates and progressively enhanced answer quality—underscoring its adaptability and superior reasoning depth.

**FERA-Q.** FERA-Q corresponds to a simplified setting where the LLM is limited to predicting only the final answer to each question, without generating intermediate reasoning steps. In this setup, when clients select in-context demonstrations from their local datasets, they discard the reasoning trajectories $S_{\{1:T\}}$ and retain only the final answers $a$. This setting reflects scenarios in which the LLM lacks the ability to handle or benefit from step-by-step reasoning. During inference, each client uses the selected final-answer-only demonstrations to generate predictions for the server-issued query set. The predicted answers are then sent back to the server. The server performs aggregation using the *Uncertainty-Aware Weighted Averaging* (UA-WA) strategy, which weights client responses based on their confidence scores. The aggregated results are used to update the global query–answer set, enabling iterative refinement even without explicit reasoning steps.

**FERA-Free.** FERA-Free refers to a setting where clients have local LLMs but no labeled data—only question prompts are available locally. Since clients cannot construct in-context demonstrations from their own data, they rely entirely on query-answer pairs provided by the server. In each round, the server distributes example queries and answers. Clients use these examples to prompt their local LLMs and generate responses to new server queries. The server then aggregates re-

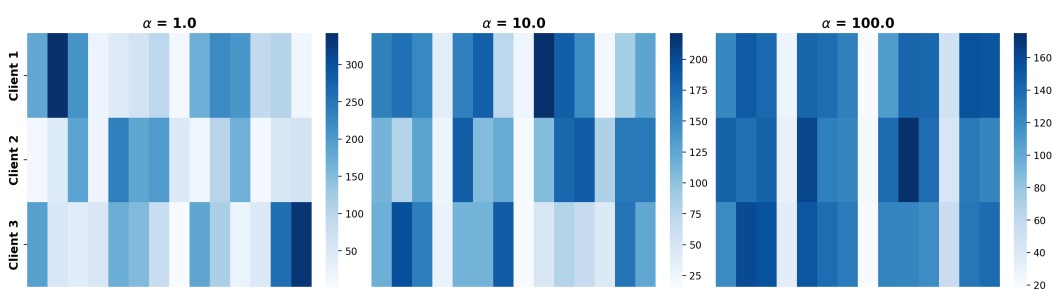

Figure 7: Client dataset distribution under different $\alpha$ settings on the MMLU-Pro benchmark.

sponses using uncertainty-aware methods (UA-WA or UA-SCA) and refines the query-answer set for the next round. This setup captures a practical constraint where local data may be unlabeled due to privacy, cost, or domain limitations. FERA-Free shows that meaningful collaboration is possible even without local supervision by leveraging server-side guidance.

# E EXPERIMENT SETUP

## E.1 CONSTRUCTION OF CLIENT DATASETS

To evaluate FERA under realistic heterogeneous and domain-specialized conditions, we simulate a federated environment by partitioning each benchmark into client-specific subsets.

**MMLU-Pro.** MMLU-Pro spans 14 diverse subject areas—including physics, medicine, philosophy, economics, and law—which naturally provides a multi-domain foundation for studying specialized client expertise. To introduce controlled heterogeneity across clients, we construct client partitions using a Dirichlet-based sampling scheme. Specifically, for each client, we draw a label-distribution vector $q \in \mathbb{R}^M$ from a Dirichlet distribution $q \sim \mathrm{Dir}(\alpha p)$, where $p$ denotes the global class distribution and $\alpha$ controls the degree of non-IID variation. Smaller values of $\alpha$ yield more skewed, domain-focused partitions (i.e., clients specializing in a small subset of subjects), whereas larger values produce more balanced distributions. We consider three heterogeneity levels with $\alpha \in \{1.0, 10, 100\}$, covering a wide spectrum from highly specialized to near-homogeneous splits. Examples of the resulting client-domain distributions are shown in Figure 7.

**AQUA-RAT and GSM8K.** These datasets do not provide explicit category labels. In line with prior work, we partition them using random splits across clients. Although these tasks do not include domain metadata, the federated setting still benefits from distributional diversity introduced by varying client sample sizes and reasoning styles.

## E.2 IMPLEMENTATION DETAILS

For all experiments involving FERA and its variants, we fix the number of clients to three. We evaluate these algorithms on the MMLU-Pro, GSM8K, and AQUA-RAT benchmarks, covering both reasoning and standard QA tasks. On the client side, we use Qwen3-4B and Llama3.1-8B as base models. During prediction, we set the sampling parameters to $top\_p = 0.8$ and temperature $= 0.3$. On the server side, no LLM is deployed for standard QA tasks. For complex reasoning tasks, the default server model is set to GPT-4o-mini.

For LLM-Debate Du et al. (2023), we ensure a fair comparison with FERA by adopting the same client models and the same number of clients as in the FERA setup. Moreover, the summarization model in LLM-Debate is configured to match the server model used in FERA, and the number of debate rounds is set to 5.

For FedAvg (McMahan et al., 2017), We implement FedAvg following the OpenFedLLM framework Ye et al. (2024) on two LLMs: `Llama-3.1-8B-Instruct` for the **MMLU-Pro** benchmark and `Mistral-7B-Instruct-v0.2` for **GSM8K**, ensuring consistency with other baselines. The training process consists of 50 communication rounds involving three clients, with data partitioned

according to to a Dirichlet distribution. Each client fine-tunes the model locally using a batch size of 16, a sequence length of 256, and one gradient accumulation step, with a learning rate of 1e-5. To improve parameter efficiency, Low-Rank Adaptation (LoRA) is applied and 8-bit quantization Hu et al. (2022). After each local update, model weights are aggregated using FedAvg on a central server, which then redistributes the updated global model to clients for the next round of local fine-tuning.

For FLora (Wang et al., 2024c), we conduct experiments using `Llama-3-8B-Instruct` Touvron et al. (2023a) and `Qwen3-4B` Yang & Team (2025) in a heterogeneous setting, where clients are assigned different LoRA rank values—i.e., [64, 32, 16]—following the default configuration in their official implementations. All hyperparameters are kept unchanged, except for the number of clients. The models are trained using our client data and evaluated on our server data.

### E.3  COMMUNICATION COST

Transmission cost refers to the communication overhead between clients and the server, measured in total bits exchanged across different algorithms. For FERA, its variants, and LLM-Debate, we set the number of iterative update rounds to six. In contrast, for traditional federated learning baselines such as FLora and FedAvg, the number of communication rounds is set to three.

Both questions and responses are tokenized, with each response capped at a maximum of 256 tokens. The total number of queries is fixed at 70 across all benchmarks. Transmission cost accounts for both the tokens sent by clients and the server in each round, allowing for a fair comparison of communication efficiency across different methods.

### E.4  DEMONSTRATION SELECTION

In **Step 2** and **Step 3** of Algorithm 1, FERA selects in-context demonstrations either from the server's query set or from the local client dataset. These demonstrations are used to facilitate client-side labeling and to update the server's query–answer set. The selection process is governed by Equation 3.1 and Equation 3.2, which implement a similarity–diversity trade-off strategy. In both equations, the function Sim measures the similarity between reasoning examples. To compute this, each example is first embedded using the `paraphrase-MiniLM-L6-v2` model (Reimers, 2019), which converts the textual reasoning into fixed-length vectors. Cosine similarity is then applied by default to quantify the similarity between embedded representations. The function Div captures the diversity within the selected set of demonstrations, encouraging the inclusion of examples that are distinct from one another. This prevents redundancy and promotes a richer representation of reasoning styles. A trade-off exists between selecting highly relevant demonstrations (those similar to the target query) and ensuring sufficient diversity within the set. This trade-off is controlled by the hyperparameter $\lambda$, which balances the weight between relevance and diversity. Unless otherwise specified, we set $\lambda = 0.5$ as the default value.

### E.5  UNCERTAINTY CALCULATION

We use the uncertainty scores to guide the aggregation of client responses on the server side. Each client generates both predictions and their corresponding logits, from which we calculate uncertainty following the approach of Duan et al. (2023); Farquhar et al. (2024); Zhang et al. (2025). To quantify the model's confidence in its generated responses, we employ a token-level entropy-based uncertainty estimation approach. Specifically, for each generated token position $t$, we calculate the entropy of the probability distribution over the vocabulary:

$$H_t = -\sum_{i=1}^{V} p_{t,i} \cdot \log(p_{t,i} + \varepsilon), \tag{E.1}$$

where $p_{t,i}$ represents the softmax probability of token $i$ at position $t$, $V$ is the vocabulary size, and $\varepsilon = 1 \times 10^{-10}$ ensures numerical stability. The overall uncertainty score is computed as the average entropy across all $T$ generated tokens:

$$U = \frac{1}{T} \sum_{t=1}^{T} H_t. \tag{E.2}$$

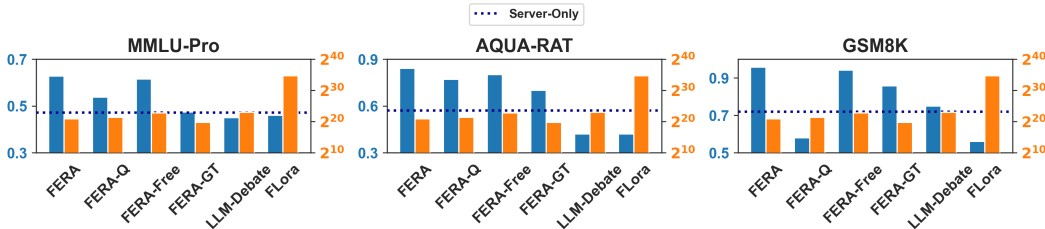

Figure 8: Performance comparison of FERA and its variants against other baselines across three benchmarks: MMLU-Pro, AQUA-RAT, and GSM8K. For MMLU-Pro, the Dirichlet concentration parameter is set to $\alpha = 10.0$ to simulate moderate client-level data heterogeneity. All results are obtained using Qwen3-4B as the client model.

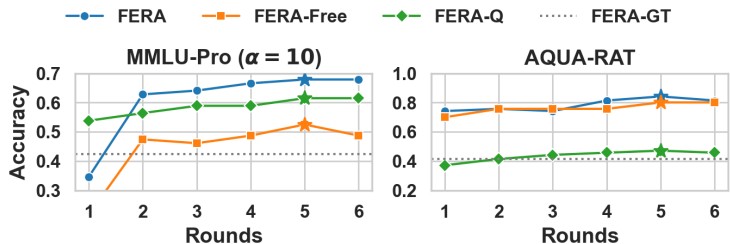

Figure 9: Effect of interaction round count on the performance of FERA and FERA-Free in the MMLU-Pro benchmark. The Dirichlet concentration parameter is set to $\alpha = 10.0$ to simulate moderate client-level data heterogeneity. All experiments use Qwen3-4B as the client model.

This metric captures the model's predictive confidence, where higher entropy values indicate greater uncertainty in token selection, while lower entropy values reflect more confident probability distributions. This approach provides a straightforward yet effective measure of generation uncertainty that can be computed directly from the model's output logits without requiring additional training or calibration, making it well-suited for our federated learning framework.

## F SUPPLEMENTARY EXPERIMENTS

### F.1 MAIN RESULTS USING QWEN3-4B AS THE CLIENT MODEL

Figure 8 presents the performance of FERA using Qwen3-4B as the client model, in comparison with other baselines and FERA variants. The results demonstrate that even with Qwen3-4B, FERA consistently outperforms competing methods, highlighting its robustness across different client models.

Figure 9 illustrates how the performance of FERA, FERA-Free, and FERA-Q evolves with an increasing number of interaction rounds. As the number of iterations increases, all variants show noticeable performance gains, demonstrating the effectiveness of the iterative refinement mechanism in the proposed framework.

### F.2 ADDITIONAL ABLATION STUDY

**Effect of Specialized Domains.** To evaluate FERA in settings where client expertise is concentrated within a particular domain, we conduct an additional experiment using the `law` category of MMLU-Pro, which is the domain where the server model shows the weakest standalone performance. We construct a server-side evaluation set by selecting 50 law questions, and distribute all remaining MMLU-Pro questions across clients using the same non-IID Dirichlet partitioning as in our main experiments. This yields a domain-imbalanced configuration in which the server has limited direct exposure to law-domain data and must rely on clients that hold the majority of the relevant information. As shown in Figure 11, FERA steadily improves over communication rounds

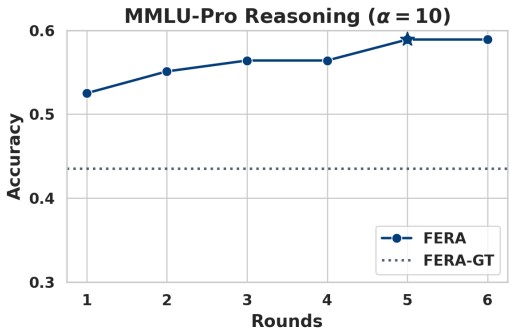
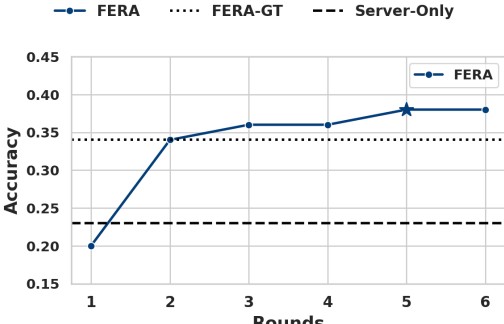

Figure 10: FERA performance on the MMLU-Pro reasoning benchmark. Both the client and server models are GPT-4o-mini.

Figure 11: FERA performance in a specialized-domain setting on the MMLU-Pro law category. Client models use Qwen3-4B, and the server model is GPT-4o-mini.

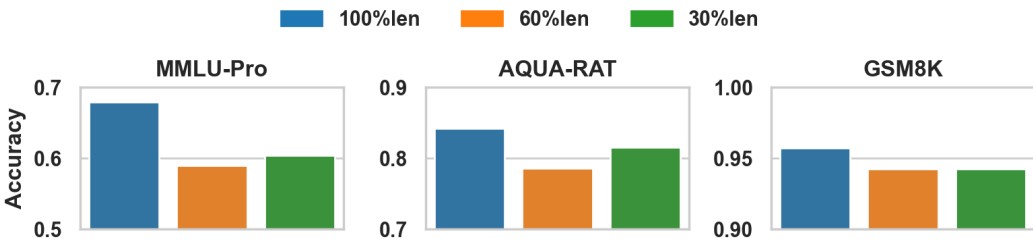

Figure 12: Performance of FERA under Varying Client Reasoning Response Quality.

and outperforms all baselines, indicating that the framework can effectively integrate specialized client knowledge even when the server begins with weak domain proficiency.

**Effect of Client Model Capacity.** In our default configuration, clients run open-source LLMs (e.g., Qwen3-4B or Llama-3.1-8B), while the server model varies by task. In this ablation, to isolate the impact of using a closed-source stack and to remove model heterogeneity between endpoints, we set both the client and the server to GPT-4o-mini. The GPT-4o-mini API exposes response-level scores (log-probability–based confidences), which we use directly to compute uncertainty for our Uncertainty-Aware Demonstration Selection and Uncertainty-Aware Aggregation, without any additional reward/critic model. This keeps the uncertainty signal aligned with the generator's own beliefs and ensures a fair comparison under identical decoding settings. The results, presented in Figure 10, demonstrate that FERA remains effective even in a closed-source setting. It consistently outperforms the FERA-GT baseline across communication rounds, indicating that FERA is model-agnostic and capable of leveraging native confidence signals to enhance performance.

**Effect of Client Model Capacity Heterogeneity.** To assess how heterogeneous model capacities affect FERA's behavior, we conduct an ablation in which clients use LLMs of substantially different sizes: Qwen3-4B, Qwen3-1.7B, and Qwen3-0.6B. This configuration introduces significant variation in both reasoning capability and uncertainty patterns across clients. The results, presented in Figure 13, show that smaller models exhibit higher initial uncertainty, reflecting their lower predictive strength, but their uncertainty decreases steadily over communication rounds. Moreover, the overall performance of FERA remains comparable to the homogeneous Qwen3-4B setting. These findings indicate that the uncertainty-aware weighting mechanism effectively moderates less reliable client signals and preserves stable aggregation despite pronounced disparities in model scale.

**Effect of Uncertainty Characteristics.** To isolate uncertainty characteristics from model capacity, we conduct an ablation in which all clients use the same model (Qwen3-4B) but adopt different decoding temperatures of 0.3, 0.5, and 0.8. Higher temperatures introduce greater sampling vari-

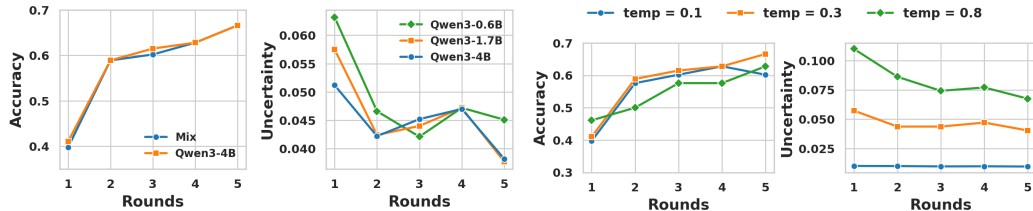

Figure 13: Effect of model-capacity heterogeneity on FERA performance for the MMLU-Pro benchmark.

Figure 14: Effect of Uncertainty Characteristics on FERA performance for the MMLU-Pro benchmark.

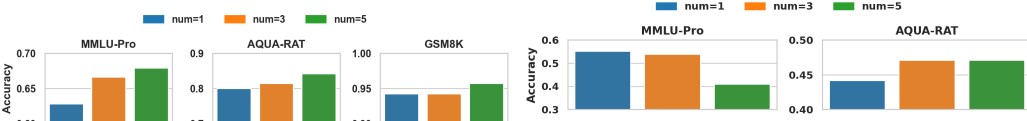

Figure 15: Effect of in-context demonstration quantity on FERA performance for differnt benchmarks.

Figure 16: Effect of the number of in-context demonstrations on FERA-Q performance for MMLU-Pro and AQUA-RAT benchmarks.

ability, which correspondingly yields larger initial uncertainty scores. The results, presented in Figure 14, show that despite these differences, FERA achieves nearly identical accuracy across all three settings, and the average uncertainty steadily decreases over communication rounds. These findings indicate that FERA's iterative refinement process effectively mitigates variability in generation behavior and converges toward stable reasoning even under increased stochasticity.

**Effect of Demonstration Quality.** We examine how client-side data quality affects the performance of FERA. To simulate degraded reasoning responses in the client data, we randomly subsample a proportion of reasoning steps from each example, thereby reducing the completeness of the provided demonstrations. For instance, the `30%len` setting retains only 30% of the original reasoning steps. Formally, let the original reasoning response be denoted as $s_{\{1:T\}} = (s_1, \ldots, s_T)$. Under the 30% setting, we sample an index set $S = \{i_1, \ldots, i_{T'}\} \subset \{1, \ldots, T\}$, with cardinality $|S| = T' = \lfloor 0.3T \rfloor$, drawn uniformly at random without replacement. The truncated reasoning response is then defined as $s'_{\{1:T'\}} = (s_{i_1}, \ldots, s_{i_{T'}})$, where the indices are ordered such that $i_1 < \cdots < i_{T'}$. As shown in Figure 12, truncating the reasoning responses used as demonstrations leads to reduced response quality, which in turn causes a noticeable decline in the overall performance of FERA.

**Effect of Demonstration Quantity.** We investigate the impact of the number of context demonstrations on FERA's performance by comparing setups with 1, 3, and 5 examples. As shown in Figure 15, 16 increasing the number of context examples improves the LLM's understanding of the query, resulting in higher response accuracy, which will also lead to the better performance of FERA and FERA-Q.

**Effect of Varying Client Numbers.** We investigate the impact of client population size by comparing FERA's performance under configurations with 3 and 5 clients. As shown in Figures 17 and 18, increasing the number of clients leads to a noticeable decline in performance. This degradation can be attributed to increased data heterogeneity across clients, which poses greater challenges for effective aggregation and weakens the overall performance of FERA and FERA-Q.

## G PRIVACY ANALYSIS OF THE FED-COT FRAMEWORK DETAILS

The main text analyzes FERA's robustness against prompt extraction attacks. In this section, we present the GPT-4o prompt used for demonstration reconstruction, along with real-world cases illustrating how FERA responds under such attacks. The results are shown in Figure 20.

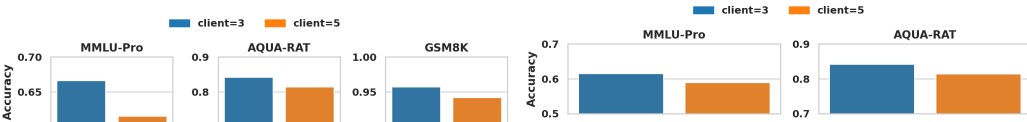

Figure 17: Effect of number of clients on FERA performance for different benchmarks.

Figure 18: Effect of the number of clients on FERA-Q performance for MMLU-Pro and AQUA-RAT benchmarks.

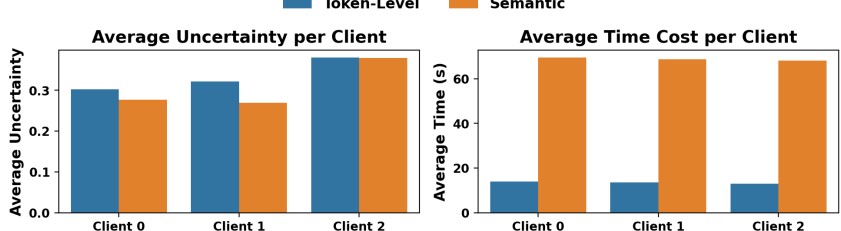

Figure 19: Comparison of token-level and semantic uncertainty. **Left**: Average uncertainty values computed for ten sampled queries across three clients. **Right:** Average computation time per client.

To further assess concerns regarding client-side privacy, particularly whether client-generated responses may inadvertently disclose private information beyond the server-issued query, we conducted additional experiments on the MMLU-Pro benchmark. Specifically, we simulated a federated setup with three clients using LLaMA3-8B-Instruct under a Dirichlet partitioning scheme with concentration parameter $\alpha = 10$. We then analyzed the generated CoTs for the presence of personal identifiers.

To quantify potential privacy leakage, we applied the "bert-base-NER" model (Devlin et al., 2019) to both the original server prompts and the client-generated CoTs. Following the methodology of Edemacu & Wu (2025), we measured how often CoTs contained personal identifiers not already present in the original prompts. The results, summarized in Table 1, show that only a small fraction of CoTs included such identifiers, indicating that FERA poses minimal risk of unintended privacy leakage in this setting.

# H    DESIGN CONSIDERATIONS FOR THE UNCERTAINTY MEASURE

In the federated reasoning setting, clients operate under strict privacy, computational, and communication constraints, which make many uncertainty-estimation techniques used in centralized LLM deployments impractical. Approaches such as semantic uncertainty, model-specific confidence scores, or auxiliary verifier–based methods generally require multiple forward passes, additional embedding or classifier modules, or model-dependent calibration procedures. These operations introduce substantial inference overhead, increase communication cost, and conflict with FERA's training-free and model-agnostic design objectives. Token-level entropy, by contrast, offers a lightweight and consistent alternative. It is computed directly from the token-probability distributions already produced during decoding, requires no auxiliary components or calibration, and adds no additional computational or architectural assumptions for the client.

To evaluate whether more complex alternatives offer meaningful advantages, we conducted a small comparative study. Ten questions were sampled from the server's evaluation set, and for each of three clients we computed both token-level entropy and semantic-uncertainty scores, while also recording the computation time required for each method. The token-level uncertainty score is computed as described in Section E.5, and the semantic-uncertainty measure follows the formulation in Kuhn et al. (2023). The results, summarized in Figure 19, show that the two uncertainty measures are closely aligned in magnitude across all clients, indicating that semantic uncertainty does not yield qualitatively different signals. In contrast, its computational cost is substantially higher—typically 4–5× slower—due to additional encoding and comparison steps. These observations reinforce our

design choice: token-level entropy provides a reliable, efficient, and privacy-compatible uncertainty signal that aligns with the constraints and objectives of federated reasoning systems like FERA.

**GPT-4o Prompt for Reconstruct ICL Examples**

```
response = openai.ChatCompletion.create(
  model="gpt-4o",
  messages=[
    {
      "role":  "system",
      "content":  "You are a helpful assistant that specializes in
natural language generation and in-context learning design."
    },
    {
      "role":  "user",
      "content":  (
        "I am using in-context learning to answer a question, where
the context is composed of several relevant question--CoT answer
pairs.  I will provide the query and its corresponding final
answer.Please assist in reconstructing a set of five question-CoT
answer pairs that are topically and logically aligned with the
final answer, serving as effective context examples."

        "Question:  XXXXX"
        "Answer:  XXXXX"
      )
    }
  ]
)
```

**Server Query**

Natalia sold clips to 48 of her friends in April, and then she sold half as many clips in May. How many clips did Natalia sell altogether in April and May?

**Client Response**

In April, Natalia sold clips to 48 of her friends, which means she sold a total of 48 clips in that month.
In May, she sold half as many clips as she did in April.
Therefore, the number of clips she sold in May is calculated as follows: 48 divided by 2 equals 24 clips.
To find the total number of clips sold by Natalia in both months, we add the clips sold in April and May together: 48 clips from April plus 24 clips from May equals 72 clips in total.
Thus, the consistent reasoning across all the provided answers confirms that Natalia sold a total of 72 clips in April and May combined.
The answer is 72.
The final answer is 72.

**Client Provide Context Examples**

**Q1:** Sally is selling boxes of crackers for her scout troop's fund-raiser. If she sells 50% more on Sunday than she sold on Saturday, then she'll have sold a total of 150 boxes on the two days. How many boxes did she sell on Saturday?
**Answer1:** Let S be the number of boxes sold on Saturday.
S+1.5S=150
2.5S=150
S=<<60=60>>60
Sally sold 60 boxes on Saturday.
The answer is: 60
**Q2:** Ginger owns a flower shop, where she sells roses, lilacs, and gardenias. On Tuesday, she sold three times more roses than lilacs, and half as many gardenias as lilacs. If she sold 10 lilacs, what is the total number of flowers sold on Tuesday?
**Answer2:** With 10 lilacs sold, three times more roses than lilacs is 3*10=<<10*3=30>>30 roses.
Half as many gardenias as lilacs is 10/2=<<10/2=5>>5 gardenias.
In total, there were 10+30+5=<<10+30+5=45>>45 flowers sold on Tuesday.
The answer is: 45
**Q3:** Allison went to the craft store with her friend Marie to buy some glue sticks and construction paper. Allison bought 8 more glue sticks than Marie, but Marie bought six times as many packs of construction paper as Allison. If Marie bought 15 glue sticks and 30 packs of construction paper how many craft supply items did Allison buy all together?
**Answer3:** Allison bought 8+15=<<8+15=23>>23 glue sticks
Allison bought 30/6=<<30/6=5>>5 packs of construction paper.
Allison bought 23+5= <<23+5=28>>28 craft supply items
The answer is: 28

**Reconstruct Client Context Examples**

**Q:** Jason collected 36 stamps in January. In February, he collected twice as many stamps as in January. How many stamps did Jason collect in total during January and February?
**Answer:** In January, Jason collected 36 stamps.
In February, he collected twice as many as he did in January.
So, the number of stamps in February is 36 × 2 = 72.
To find the total, we add both months together: 36 + 72 = 108.
The answer is 108.
**Q:** Mia read 30 pages on Monday. On Tuesday, she read 10 fewer pages than she did on Monday. How many pages did Mia read on Monday and Tuesday combined?
**Answer:** On Monday, Mia read 30 pages.
On Tuesday, she read 10 fewer pages than Monday, so 30 − 10 = 20 pages.
To find the total pages read, add both days: 30 + 20 = 50 pages.
The answer is 50.
**Q:** Liam had 60 marbles. He gave one-third of them to his friend. How many marbles does Liam have left now?
**Answer:** Liam started with 60 marbles.
He gave away one-third, which is 60 ÷ 3 = 20 marbles.
To find how many he has left, subtract: 60 − 20 = 40 marbles.
The answer is 40.

Figure 20: Privacy Analysis

|  | Client 1 | Client 2 | Client 3 | Avg. |
|---|---|---|---|---|
| Value (%) | 0.031 | 0.049 | 0.051 | 0.043 |

Table 1: Fraction of client-generated reasoning response containing personal identifiers, as detected by the `bert-base-NER` model.

# I    CASE STUDY

In this section, we present real examples that illustrate how CoT answers are updated across interaction rounds. These cases are shown in from Figure 21 to Figure 23.

**Server Query1**

Gerald spends $100 a month on baseball supplies. His season is 4 months long. He wants to use the months he's not playing baseball to save up by raking, shoveling, and mowing lawns. He charges $10 for each. How many chores does he need to average a month to save up for his supplies?

**Round1**

Gerald spends $100 a month on baseball supplies, and his baseball season lasts for 4 months.
Therefore, he needs to save up a total of $400 for the entire season (4 months x $100/month).
During the off-season, which is also 4 months long, he wants to save up for these supplies by doing chores.
He charges $10 for each chore he completes.
To determine how many chores he needs to do each month to save up for his supplies, we first calculate how much he needs to earn each month during the off-season.
Since he needs to save $100 each month for his supplies, we can find out how many chores he needs to do by dividing the amount he needs to save by the amount he earns per chore:

$$
\text{Chores needed per month} = \frac{\text{Amount needed per month}}{\text{Earnings per chore}} = \frac{100}{10} = 10
$$

Thus, Gerald needs to complete 10 chores each month during the off-season to save up for his baseball supplies.
The answer is 10.
The final answer is 10.

**Round2**

Gerald spends $100 a month on baseball supplies, and his baseball season lasts for 4 months.
Therefore, he will not be playing baseball for the remaining 8 months of the year.
To save up for his supplies during these 8 months, he needs to accumulate a total of $100 per month for 8 months, which amounts to $100 x 8 = $800.
Gerald charges $10 for each chore he completes, so to determine how many chores he needs to do to save up the required $800, we can divide the total amount he needs by the amount he earns per chore.
This gives us $800 / $10 = 80 chores.
Thus, Gerald needs to average 80 chores over the 8 months to save up for his baseball supplies.
The answer is 80.
The final answer is 80.

**Round3**

Gerald spends $100 a month on baseball supplies, and since his season lasts for 4 months, he needs to save up a total of $100 multiplied by 4, which equals $400 for the entire season.
To earn this amount, he plans to do chores, charging $10 for each one.
Therefore, to find out how many chores he needs to complete, we divide the total amount he needs to save, $400, by the amount he earns per chore, $10.
This calculation gives us $400 divided by $10, which equals 40.
Thus, Gerald needs to average 40 chores a month to save up for his baseball supplies.
The answer is 40.
The final answer is 40.

**Round4**

Gerald spends $100 a month on baseball supplies, and his season lasts for 4 months.
Therefore, the total amount he needs for the season is $100 x 4 = $400.
After the season, he wants to save up for the next season during the 8 months he is not playing baseball.
To find out how much he needs to save during these 8 months, we can calculate the total amount he needs for the next season, which is again $400.
Now, since he wants to save this amount during the 8 months, he needs to save $400 over those months.
To determine how much he needs to save each month, we divide the total amount by the number of months: $400 / 8 = $50 per month.
Gerald charges $10 for each chore he does.
To find out how many chores he needs to complete each month to save $50, we divide the amount he needs to save by the amount he earns per chore: $50 / $10 = 5 chores.
Thus, Gerald needs to average 5 chores a month to save up for his supplies.
The answer is 5.

Figure 21: Example Reasoning Answer iterative update for Query 1

**Server Query2**

Angela is a bike messenger in New York. She needs to deliver 8 times as many packages as meals. If she needs to deliver 27 meals and packages combined, how many meals does she deliver?

**Round1**

To solve the problem, we need to define the variables for meals and packages.
Let $m$ represent the number of meals and $p$ represent the number of packages.
According to the problem, Angela needs to deliver 8 times as many packages as meals, which gives us the equation:
$$m = 8p$$
We also know that the total number of meals and packages combined is 27:
$$m + p = 27$$
Now we can substitute the expression for $m$ from the first equation into the second equation:
$$8p + p = 27$$
Combining like terms, we have:
$$9p = 27$$
Next, we divide both sides by 9 to solve for $p$:
$$p = 3$$
Now that we have the number of packages, we can find the number of meals by substituting $p$ back into the equation for $m$:
$$m = 8p = 8 \times 3 = 24$$
Thus, Angela delivers 24 meals.

**Round2**

Let $x$ be the number of meals and $y$ be the number of packages.
According to the problem, Angela needs to deliver 8 times as many packages as meals, which gives us the equation:
$$x = 8y$$
Additionally, we know that the total number of meals and packages combined is 27, leading to the equation:
$$x + y = 27$$
Now, we can substitute the expression for $x$ from the first equation into the second equation:
$$8y + y = 27$$
Combining like terms, we have:
$$9y = 27$$
Next, we divide both sides by 9 to solve for $y$:
$$y = 3$$
Now that we have the number of packages, we can substitute $y$ back into the equation for $x$:
$$x = 8y = 8(3) = 24$$
Thus, Angela delivers 24 meals.

**Round3**

Let m be the number of meals and p be the number of packages.
We know from the problem that Angela needs to deliver a total of 27 meals and packages combined, which gives us the equation:
m + p = 27
Additionally, we know that she needs to deliver 8 times as many packages as meals, leading to the equation:
p = 8m
Now, we can substitute the second equation into the first equation.
By replacing p in the first equation with 8m, we get:
m + 8m = 27
Combining like terms results in:
9m = 27
To find the number of meals (m), we divide both sides by 9:
m = 27 / 9
m = 3
Now that we have the number of meals, we can find the number of packages by substituting m back into the equation for p:
p = 8m
p = 8 * 3
p = 24
Thus, Angela delivers 3 meals and 24 packages, which satisfies both conditions given in the problem.

Figure 22: Example Reasoning Answer iterative update for Query 2

**Server Query3**

Sansa is a famous artist, she can draw a portrait and sell it according to its size. She sells an 8-inch portrait for $5, and a 16-inch portrait for twice the price of the 8-inch portrait. If she sells three 8-inch portraits and five 16-inch portraits per day, how many does she earns every 3 days?

**Round1**

To determine Sansa's earnings over three days, we first need to establish the price of each portrait size.
The price of an 8-inch portrait is given as $5.
Since a 16-inch portrait costs twice as much as an 8-inch portrait, we calculate the price of the 16-inch portrait as follows:
Price of a 16-inch portrait = $5 * 2 = $10.
Next, we calculate Sansa's daily earnings from selling the portraits.
She sells three 8-inch portraits per day, so her earnings from these portraits are:
Earnings from 8-inch portraits = 3 * $5 = $15.
She also sells five 16-inch portraits per day, which brings her earnings from these portraits to:
Earnings from 16-inch portraits = 5 * $10 = $50.
Now, we can find her total daily earnings by adding the earnings from both types of portraits:
Total daily earnings = Earnings from 8-inch portraits + Earnings from 16-inch portraits = $15 + $50 = $65.
To find her total earnings over three days, we multiply her daily earnings by 3:
Total earnings over 3 days = $65 * 3 = $195.
Thus, after consolidating the reasoning
The final answer is 210.

**Round2**

To determine Sansa's earnings over a period of 3 days, we first need to calculate the prices of the portraits she sells.
**Price of Portraits**:
  - An 8-inch portrait is sold for $5.
- A 16-inch portrait is sold for twice the price of the 8-inch portrait, which is $5 * 2 = $10.
**Daily Sales**:
  - Sansa sells 3 8-inch portraits per day.
Therefore, her earnings from the 8-inch portraits per day are:
  $[3 \text{ portraits} \times \$5 = \$15\]$
  - She also sells 5 16-inch portraits per day.
Thus, her earnings from the 16-inch portraits per day are:
  $[5 \text{ portraits} \times \$10 = \$50\]$
3.**Total Daily Earnings**:
  - To find her total earnings per day, we add the earnings from both types of portraits:
  $[ \$15 + \$50 = \$65\]$
4.
**Earnings Over 3 Days**:
  - To find her total earnings over 3 days,
The final answer is 200.

**Round3**

To solve the problem, we start by determining the price of the 16-inch portrait.
Since Sansa sells an 8-inch portrait for $5, the price of a 16-inch portrait, which is twice the price of the 8-inch portrait, is calculated as follows:
Price of 16-inch portrait = $5 * 2 = $10.
Next, we calculate the daily revenue from selling the portraits.
Sansa sells 3 8-inch portraits and 5 16-inch portraits each day.
First, we find the revenue from the 8-inch portraits:
Revenue from 3 8-inch portraits = 3 * $5 = $15.
Now, we calculate the revenue from the 16-inch portraits:
Revenue from 5 16-inch portraits = 5 * $10 = $50.
Now, we can find the total daily revenue by adding the revenue from both types of portraits:
Total daily revenue = $15 + $50 = $65.
To find out how much Sansa earns over 3 days, we multiply the daily revenue by 3:
Revenue earned in 3 days = $65 * 3 = $195.
Thus, after consolidating all the reasoning, we conclude that Sansa earns a total of $
The final answer is 195.

Figure 23: Example Reasoning Answer iterative update for Query 3

