# OpenReview forum: "FERA: Uncertainty-aware Federated Reasoning for Large Language Models"
_ICLR.cc/2026/Conference — Submitted to ICLR 2026_

### Official Review · Reviewer_R7vv · 2025-10-28

**Soundness:** 2
**Presentation:** 3
**Contribution:** 2
**Rating:** 2
**Confidence:** 3

**Summary:**

Authors claim that the rLLM development is hard in real deployment setting due to privacy issue, and the Federated methods come with high computation and communication cost. To address these issues, they propose a parameter-free federated reasoning framework (FERA). In FERA, the server distributes queries to clients, which generate reasoning steps and answers using their private data. These reasoning steps are added to each client’s dataset and used to refine subsequent predictions. To handle data heterogeneity, the authors introduce two aggregation mechanisms (UA-WA and UA-SCA), which weight client contributions based on confidence and reasoning consistency.

**Strengths:**

1.	The concept of a parameter-free federated reasoning framework is novel and remains relatively underexplored in the current literature.

2.	The paper is well-written, with clearly stated motivations and challenges. The proposed contributions align well with these challenges, and the overall structure is coherent.

**Weaknesses:**

1.	As mentioned earlier, the concept of FERA is novel and intuitively aligns with human collaborative reasoning. However, it does not yet appear robust enough for practical deployment, as the reasoning traces generated by LLMs are inherently noisy. Existing LLM documentation explicitly states that “models generate text token by token based on patterns in data, not by performing deliberate reasoning or internal planning.” Thus, the reasoning steps produced by local LLMs should be viewed as patterns learn from training rather than faithful representations of actual reasoning processes.

2.	Even if we assume that the reasoning steps reflect the model’s real thinking process, the proposed aggregation mechanism still faces a critical issue. The central goal of FERA claimed is to aggregate reasoning, not to re-reason from scratch. However, during aggregation, the server’s LLM summarizes all client reasoning patterns to extract a dominant one, which may easily injects its own reasoning bias into the process. Therefore, the final aggregated reasoning trace may no longer represent the authentic reasoning patterns of the clients.

3.	The paper lacks a detailed system overview diagram. Figure 1 is insufficient to fully convey the workflow and interactions within the proposed framework.

4.	The evaluation is not comprehensive. Additional comparisons with established federated learning and parameter-free methods are needed, and there is no assessment of the truthfulness or faithfulness of the generated reasoning steps.

**Questions:**

1.	How do the authors ensure that the reasoning steps generated by LLMs are reliable, given that these steps may not reflect true reasoning but only text patterns?

2.	How does the server prevent its summarization process from adding its own bias, so that the final reasoning truly represents the clients’ reasoning rather than the server’s interpretation?

---

> ### Author Response · Authors · 2025-11-22
> **To Reviewer R7vv**
>
> Thanks for your helpful feedback.
>
> **Q1:** As mentioned earlier, the concept of FERA is novel and intuitively aligns with human collaborative reasoning. However, it does not yet appear robust enough for practical deployment, as the reasoning traces generated by LLMs are inherently noisy. Existing LLM documentation explicitly states that “models generate text token by token based on patterns in data, not by performing deliberate reasoning or internal planning.” Thus, the reasoning steps produced by local LLMs should be viewed as patterns learn from training rather than faithful representations of actual reasoning processes.
>
> **A1:** We agree that the reasoning traces produced by current LLMs may be noisy and not fully faithful representations of internal reasoning. However, this limitation is not unique to our setting—it applies broadly to widely used reasoning frameworks such as Chain-of-Thought [1], Self-Consistency [2], and Debate [3], all of which rely on model-generated intermediate steps rather than verifiable logical derivations. Despite these approaches remain effective in practice because their intermediate outputs still encode useful information, even if they are not perfectly faithful representations of reasoning [1-5].
>
> FERA follows this same principle but introduces an additional safeguard. The framework does not assume that every reasoning trace is accurate; instead, it explicitly quantifies the model’s uncertainty over its own reasoning. Noisy or unstable reasoning naturally produces higher uncertainty and is therefore down-weighted in the aggregation step. This design makes FERA robust to imperfect reasoning and prevents the aggregation from being dominated by unreliable traces. Our experiments empirically confirm that FERA remains stable even when local reasoning quality varies significantly in ***Section F.2: Effect of Demonstration Quality, p.35***. This demonstrates that the framework relies not on the absolute correctness of reasoning steps, but on the relative uncertainty structure they produce.
>
> **Q2:** Even if we assume that the reasoning steps reflect the model’s real thinking process, the proposed aggregation mechanism still faces a critical issue. The central goal of FERA claimed is to aggregate reasoning, not to re-reason from scratch. However, during aggregation, the server’s LLM summarizes all client reasoning patterns to extract a dominant one, which may easily injects its own reasoning bias into the process. Therefore, the final aggregated reasoning trace may no longer represent the authentic reasoning patterns of the clients.
>
> **A2:** Our goal in FERA is not to preserve a verbatim record of each client’s reasoning trace, but to **enhance** the server model’s prediction quality by leveraging the diverse reasoning signals provided by clients. During aggregration, the server-side LLM may introduce its own inductive bias, but this is expected and aligned with the objective: the server synthesizes multiple noisy reasoning paths into a more coherent and broadly supported rationale. Empirically, FERA delivers **substantial improvements** over the **Server-Only** baseline as shown in ***Figure 2, p.8***. This demonstrates that the aggregation mechanism successfully distills meaningful structure from client reasoning traces, even when the merged output does not exactly replicate any individual client’s reasoning.
>
> **Q3:** The paper lacks a detailed system overview diagram. Figure 1 is insufficient to fully convey the workflow and interactions within the proposed framework.
>
> **A3:** Thank you for pointing this out. Figure 1 is intentionally kept at a high level to give a clear overview of the main components of FERA. More detailed explanations of each step are provided in ***Section 3, p.3*** and the full workflow, including the client–server interactions, is described explicitly in ***Algorithm 1, p.4***.
>
> **Q4:** The evaluation is not comprehensive. Additional comparisons with established federated learning and parameter-free methods are needed. There is no assessment of the truthfulness or faithfulness of the generated reasoning steps.
>
> **A4:** Our evaluation setup follows the protocol used in [4-10], which has become a standard choice in recent work. If there are specific metrics or procedures you believe would better capture faithfulness, we would appreciate the suggestion and can include the corresponding analysis in the revision.

---

> ### Author Response · Authors · 2025-11-22
> **To Reviewer R7vv**
>
> **Q5:** How do the authors ensure that the reasoning steps generated by LLMs are reliable, given that these steps may not reflect true reasoning but only text patterns?
>
> **A5:** We agree that LLM-generated reasoning traces may be noisy or imperfect, but this limitation is shared by widely used methods such as Chain-of-Thought, Self-Consistency, and Debate, which also rely on synthetic intermediate steps rather than verifiable logic. These approaches still work well because the traces contain useful information even when not fully faithful [1-5].
>
> FERA follows the same principle but adds an explicit safeguard: it does not assume correctness of each trace and instead weights them by their uncertainty. Noisy or unstable reasoning naturally receives lower weight, preventing unreliable traces from dominating the aggregation. As shown in ***Section F.2: Effect of Demonstration Quality, p.35***, FERA remains stable even when client reasoning quality varies significantly, indicating that the framework leverages relative uncertainty patterns rather than requiring perfectly accurate reasoning steps. For more detailed analysis, please refer to **A1**.
>
> **Q6:** How does the server prevent its summarization process from adding its own bias, so that the final reasoning truly represents the clients’ reasoning rather than the server’s interpretation?
>
> **A6:** FERA is designed not to preserve each client’s reasoning verbatim, but to improve the server model by synthesizing diverse client-provided reasoning signals. Some server-side inductive bias is expected and aligns with this goal, as the server consolidates multiple noisy traces into a more coherent rationale. Empirically, FERA achieves substantial gains over the Server-Only baseline (***Figure 2, p.8***), showing that the aggregation effectively extracts useful structure even without reproducing any single client’s reasoning.
>
> ## Reference
>
> - [1] Wei, Jason, et al. "Chain-of-thought prompting elicits reasoning in large language models." Advances in neural information processing systems 35 (2022): 24824-24837.
> - [2] Wang, Xuezhi, et al. "Self-consistency improves chain of thought reasoning in language models." arXiv preprint arXiv:2203.11171 (2022).
> - [3] Du, Yilun, et al. "Improving factuality and reasoning in language models through multiagent debate." Forty-first International Conference on Machine Learning. 2023.
> - [4]Kojima, Takeshi, et al. "Large language models are zero-shot reasoners." Advances in neural information processing systems 35 (2022): 22199-22213.
> - [5]Lewkowycz, Aitor, et al. "Solving quantitative reasoning problems with language models." Advances in neural information processing systems 35 (2022): 3843-3857.
> - [6]Madaan, Aman, et al. "Self-refine: Iterative refinement with self-feedback." Advances in Neural Information Processing Systems 36 (2023): 46534-46594.
> - [7]Yao, Shunyu, et al. "React: Synergizing reasoning and acting in language models." The eleventh international conference on learning representations. 2022.
> - [8]Achiam, Josh, et al. "Gpt-4 technical report." arXiv preprint arXiv:2303.08774 (2023).
> - [9]Shao, Zhihong, et al. "Deepseekmath: Pushing the limits of mathematical reasoning in open language models." arXiv preprint arXiv:2402.03300 (2024).
> - [10]Yang, An, et al. "Qwen3 technical report." arXiv preprint arXiv:2505.09388 (2025).

---

### Official Review · Reviewer_Fjet · 2025-11-01

**Soundness:** 3
**Presentation:** 3
**Contribution:** 3
**Rating:** 6
**Confidence:** 1

**Summary:**

The paper proposes FERA (Uncertainty-Aware Federated Reasoning), a framework for performing collaborative reasoning with large language models (LLMs) in federated settings where data cannot be shared. Instead of training or fine-tuning models, FERA enables clients to perform reasoning locally and share their responses — along with uncertainty estimates — with a central server. The server then aggregates these outputs using an uncertainty-aware weighting mechanism, and further refines the reasoning through a self-critique process. This approach aims to reduce communication costs and handle data heterogeneity more effectively than traditional federated learning or simple prompting-based collaboration.

Experiments on several reasoning benchmarks show that FERA achieves better performance than standard baselines while requiring less communication. Theoretical analysis on a simplified model provides convergence insights for the uncertainty-weighted aggregation. Overall, the paper contributes a novel parameter-free federated reasoning paradigm and introduces uncertainty-aware and self-critique aggregation mechanisms. However, the approach relies on access to capable local LLMs and assumes synchronous participation, which may limit its practicality in real-world federated systems.

**Strengths:**

1. The paper proposes a novel parameter-free federated reasoning framework that enables collaboration without local model updates, balancing privacy preservation and computational efficiency.
2. The introduction of uncertainty-aware aggregation allows the system to weigh client contributions by confidence, improving robustness under heterogeneous data distributions.
3. The approach is supported by both theoretical convergence analysis and extensive experiments, providing convincing evidence of its effectiveness across multiple reasoning tasks.

**Weaknesses:**

1. I am not an expert in federated learning, so I may not be able to fully assess the technical depth of this work or identify all potential weaknesses.

**Questions:**

It would be helpful to clarify how the uncertainty estimates are computed on each client. Are they derived directly from the LLM’s output probabilities, or through an additional calibration step?

---

> ### Author Response · Authors · 2025-11-22
> **To Reviewer Fjet**
>
> Thanks for your helpful feedback.
>
> **Q1:** The approach relies on access to capable local LLMs and assumes synchronous participation, which may limit its practicality in real-world federated systems.
>
> **A1:** We would like to clarify that FERA neither requires powerful local LLMs nor depends on strict synchronous participation. First, FERA is explicitly designed to accommodate heterogeneous client capacities, including small or weaker models. Our mixed-capacity experiment (Qwen3-4B / 1.7B / 0.6B), reported in ***Section F.2: Effect of Client Model Capacity Heterogeneity, p.34***, shows that clients with limited capacity naturally produce higher uncertainty and therefore receive lower aggregation weight, while still contributing useful signal. The overall accuracy remains stable, indicating that FERA is robust even when some clients use modest models.
>
> Second, under an asynchronous setting, FERA can still function by modifying ***line 8 in Algorithm 1, p.4*** to aggregate only the clients whose updates arrive at the current time. We acknowledge that a fully asynchronous variant would require a redesigned aggregation rule that jointly handles uncertainty and staleness. Because the main goal of this paper is to introduce a training-free federated framework that incorporates client-side dataset information into server decision making, we view the asynchronous extension as an orthogonal direction and leave it to future work. We have added ***Remark 3.6, p.6*** in the revised PDF to highlight these practical considerations.
>
>
> **Q2:** It would be helpful to clarify how the uncertainty estimates are computed on each client. Are they derived directly from the LLM’s output probabilities, or through an additional calibration step?
>
> **A2:** The uncertainty estimates on each client are computed directly from the LLM’s output probabilities, without any additional calibration step. This keeps the procedure fully parameter-free and consistent across clients. Further details on the uncertainty computation are provided in ***Section E.4, p.32***.

---

### Official Review · Reviewer_LBiw · 2025-11-02

**Soundness:** 2
**Presentation:** 3
**Contribution:** 2
**Rating:** 4
**Confidence:** 4

**Summary:**

This paper proposes FERA, an uncertainty-aware federated reasoning framework for LLMs that addresses the fundamental trade-off between privacy preservation and computational efficiency in distributed reasoning tasks. The framework eliminates the need for costly parameter updates through iterative server-client collaboration, where client responses are aggregated using uncertainty-weighted mechanisms. The authors provide theoretical convergence guarantees for a simplified linear attention model and demonstrate empirical improvements across MMLU-Pro, AQUA-RAT, and GSM8K benchmarks, showing that FERA achieves competitive performance while maintaining significantly lower communication overhead than training-based federated learning approaches.

**Strengths:**

1. The paper introduces an innovative training-free federated reasoning framework that effectively balances privacy, efficiency, and performance. The theoretical analysis using linear self-attention models provides solid mathematical foundations for the uncertainty-aware aggregation strategy.
2.  FERA's modular design with UA-WA for simple aggregation and UA-SCA for complex reasoning tasks, combined with detailed implementation protocols and prompts, makes the framework readily deployable while addressing real-world constraints of federated learning environments.

**Weaknesses:**

1. Given that the framework targets scenarios with private, domain-specific data across clients, the exclusive use of general-purpose benchmarks (MMLU-Pro, AQUA-RAT, GSM8K) may not adequately demonstrate FERA's effectiveness in specialized domains where clients possess unique expertise that pre-trained models lack.
2. While the paper considers different client models (Qwen3-4B, LLaMA-3.1-8B), it lacks thorough investigation of how varying model capabilities and their inherent uncertainty characteristics affect aggregation quality, particularly when mixing models with substantially different scales or architectures.
3. The paper mentions using token-level entropy-based uncertainty (Equation E.1-E.2) but provides limited justification for this choice over alternatives like semantic uncertainty or model-specific confidence scores, and doesn't discuss calibration issues that could affect aggregation weights.

**Questions:**

1. How does FERA perform when clients have highly specialized domain knowledge not present in pre-trained models?
2. What is the sensitivity of the aggregation mechanism to uncertainty calibration across heterogeneous models?
3. How exactly is the uncertainty score in Equation 3.3 computed in practice? The main paper refers to temperature-scaled softmax but lacks implementation details.
4. How does FERA handle adversarial or low-quality clients? The uncertainty weighting assumes honest clients, but what mechanisms prevent malicious clients from manipulating uncertainty scores to gain disproportionate influence in aggregation?

---

> ### Author Response · Authors · 2025-11-22
> **To Reviewer LBiw**
>
> Thanks for your helpful feedback.
>
> **Q1:** My biggest concern is the evaluation setup. They claim this is for private, domain-specific data, but then only test on general benchmarks. What about scenarios where clients have specialized knowledge that isn't in pre-trained models? That's the whole point of federated learning, right? This feels like a missed opportunity.
>
> **A1:** Our primary experiments focus on MMLU-Pro, AQUA-RAT, and GSM8K because these benchmarks are widely used in prior work [1-5], allowing for direct and meaningful comparison with existing baselines. At the same time, our evaluation does not rely on homogeneous data. MMLU-Pro covers 14 distinct subject areas, and our non-IID Dirichlet partitioning (controlled by the parameter $\alpha$) produces client datasets with imbalanced domain coverage that reflect realistic specialization. ***Figure 7, p.31*** in ***Section E.1, p.31*** visualizes these partitions, and ***Section 4.4, p.8*** examines how FERA behaves as domain heterogeneity increases.
>
> To further address the reviewer’s concern, we added an explicit specialized-domain ablation ***Section F.2: Effect of Specialized Domains, p.33***. Focusing on the law category—where the server model performs weakest—we construct a domain-imbalanced setting that forces the server to rely on clients holding most of the law-domain information. Across this setup, FERA shows consistent improvement over rounds and surpasses all baselines, confirming that it can effectively leverage domain-specific client expertise even when the server lacks it.
>
> Together, these results show that FERA is not limited to general-purpose benchmarks, but is well-suited for specialized, heterogeneous, and expertise-concentrated federated environments, directly addressing the reviewer’s concern.
>
> **Q2:** The heterogeneous model experiments are too limited. They test Qwen3-4B and LLaMA-3.1-8B, but what happens with bigger capability gaps? Or different architectures entirely? The aggregation might break down when mixing very different models, but we don't know.
>
> **A2:** We added ***Section F.2: Effect of Client Model Capacity Heterogeneity; Effect of Uncertainty Characteristics, p.34***, the two ablation studies that directly examine how differences in model capacity and uncertainty behavior influence the performance and stability of FERA.
>
> **(1) Mixed-model capacity setting.**
>
> We evaluate clients using Qwen3-4B, Qwen3-1.7B, and Qwen3-0.6B. Smaller models produce higher initial uncertainty, but their uncertainty decreases rapidly over rounds, and FERA’s final accuracy is close to the all–Qwen3-4B setup. This confirms that the uncertainty-aware mechanism appropriately moderates noisier client outputs and keeps aggregation stable even under substantial model-scale differences.
>
> **(2) Temperature-induced uncertainty.**
>
> To examine uncertainty independently of model size, we evaluate three settings in which all clients use Qwen3-4B but with different sampling temperatures: 0.3, 0.5, and 0.8. While higher temperatures introduce greater randomness in generation—and therefore higher initial uncertainty—FERA attains nearly identical accuracy across all configurations, with uncertainty steadily decreasing over rounds.
> These results suggest that FERA’s iterative refinement process effectively mitigates the variability introduced by higher temperatures, leading all configurations toward comparable and stable reasoning performance.
>
> These ablations demonstrate that FERA is robust to both heterogeneous model capacities and differing uncertainty characteristics, directly addressing the reviewer’s concern.
>
> **Q3:** The uncertainty estimation feels under-explored. They mention token-level entropy (Equations E.1-E.2) but don't justify why this is the right choice.
>
> **A3:** The main motivation for choosing **token-level entropy–based uncertainty** is computational efficiency. To evaluate whether alternative measures might offer a practical advantage, we conducted a small comparative study. We randomly sampled ten questions from the server’s query set, computed both token-level entropy–based uncertainty and semantic-uncertainty scores across clients, and measured their computation time. The two uncertainty metrics were highly aligned in magnitude, while semantic uncertainty [6] incurred substantially higher computational cost. This supports our choice: token-level entropy provides a lightweight and reliable uncertainty signal that remains compatible with the operational constraints of federated reasoning. The full analysis is available in ***Section E, p. 36***. We have also added ***Remark 3.5, p. 6*** to clarify the rationale behind this choice of uncertainty measure.

---

> ### Author Response · Authors · 2025-11-22
> **To Reviewer LBiw**
>
> **Q4:** What about calibration issues? Different models might be overconfident or underconfident in systematic ways. This could really mess up the aggregation weights.
>
> **A4:** We assume you are suggesting a discussion about how uncertainty could calibrate the aggregation weights. In fact, in Eq. (3.3), our aggregation weight $w_{k,m}^i$ directly depends on the uncertainty $u_{k+1, m}^i$. For any clients whose uncertainty is large, FEAR assigns less weight to the response from this client. We have made it clear in the added ***Remark 3.4, p.5***.
>
> **Q5:** Have you tested this on actual domain-specific tasks? I'm curious how it performs when clients genuinely have unique expertise.
>
> **A5:** Our use of MMLU-Pro, AQUA-RAT, and GSM8K follows prior work [1–5] and enables fair baseline comparison. The setup is not homogeneous: MMLU-Pro spans 14 subjects, and our non-IID Dirichlet partitioning creates domain-specialized client datasets (see ***Figure 7, p.31*** and ***Section 4.4, p.8***).
>
> To directly address the concern about specialized domains, we added an ablation in ***Section F.2, p.33*** focusing on the law category—the server’s weakest area. In this domain-imbalanced setting, FERA consistently improves and outperforms all baselines.
>
> Overall, the results confirm that FERA operates effectively in heterogeneous, domain-specialized federated environments. Further analysis and supporting evidence are provided in **A1**.
>
> **Q6:** How robust is the aggregation to poorly calibrated models? Some LLMs are notoriously overconfident.
>
> **A6:** We directly examined the scenario raised by evaluating FERA with heterogeneous client models(Qwen3-4B, 1.7B, and 0.6B). The results are reported in ***Section F.2: Effect of Client Model Capacity Heterogeneity, p.34***. Across this setting, FERA shows no degradation and  the performance continues to improve over rounds. This indicates that the aggregation mechanism remains robust despite uncertainty calibration differences introduced by model heterogeneity.
>
> **Q7:** Can you clarify the implementation of Equation 3.3? The paper mentions temperature-scaled softmax but I couldn't find the actual computation details. This seems important for reproduction.
>
> **A7:** The computation of the uncertainty score is detailed in ***Section E.4, p.32***. In practice, each client computes its uncertainty directly from the token-level probability distribution generated during decoding, using the entropy measure defined in ***Eqs. (E.1–E.2), p.32***. This requires no auxiliary models, calibration steps, or extra inference passes. The server then applies the temperature-scaled softmax in ***Eq. (3.3), p.5*** to convert these entropy values into normalized aggregation weights. The entire procedure is parameter-free and model-agnostic, allowing it to operate reliably across heterogeneous client models. We have updated the paper to clarify this computation pipeline.

---

> ### Author Response · Authors · 2025-11-22
> **To Reviewer LBiw**
>
> **Q8:** What about malicious clients? The uncertainty weighting seems to assume everyone's honest. But couldn't a client fake low uncertainty to dominate the aggregation? Any thoughts on defense mechanisms?
>
> **A8:** For low-quality clients, we assess robustness to unreliable clients in  ***Section F.2: Effect of Demonstration Quality, p.35***. Even when client demonstrations are intentionally degraded, FERA remains stable. These findings show that the aggregation mechanism is resilient to low-quality or strategically misreported uncertainty scores.
>
> For adversarial clients, although they may submit low-quality answers with artificially low reported uncertainty, FERA remains robust because: (i) uncertainty-weighted aggregation limits the influence of any single client, (ii) UA-SCA cross-examines and corrects inconsistent reasoning paths through server-side critique, and (iii) our theoretical analysis shows that noisy, high-variance contributors are provably down-weighted over iterations. Together, these mechanisms prevent confident-but-incorrect adversarial updates from meaningfully degrading the global reasoning quality.
>
> ## Reference
> - [1] Wang, Yubo, et al. "Mmlu-pro: A more robust and challenging multi-task language understanding benchmark." Advances in Neural Information Processing Systems 37 (2024): 95266-95290.
> - [2] Yang, An, et al. "Qwen3 technical report." arXiv preprint arXiv:2505.09388 (2025).
> - [3] Zhou, Denny, et al. "Least-to-most prompting enables complex reasoning in large language models." arXiv preprint arXiv:2205.10625 (2022).
> - [4] Wei, Jason, et al. "Chain-of-thought prompting elicits reasoning in large language models." Advances in neural information processing systems 35 (2022): 24824-24837.
> - [5] Shum, KaShun, Shizhe Diao, and Tong Zhang. "Automatic prompt augmentation and selection with chain-of-thought from labeled data." arXiv preprint arXiv:2302.12822 (2023).
> - [6] Kuhn, L., Gal, Y., & Farquhar, S. (2023). Semantic uncertainty: Linguistic invariances for uncertainty estimation in natural language generation. arXiv preprint arXiv:2302.09664.

---

### Author Response · Authors · 2025-11-22
**To all reviewers**

Thank you for your insightful comments. Below we summarize the main revisions made to the paper and indicate how they address the reviewers’ concerns:

- We added **Section E.1 (p.31)** detailing the construction of heterogeneous client datasets across benchmarks, and **Section F.2: Effect of Specialized Domains (p.33)** introducing an experiment evaluating FERA under domain-imbalanced, specialization-heavy conditions.
  These additions address **Reviewer LBiw – Q1, Q5**.

- We added **Section F.2: Effect of Client Model Capacity Heterogeneity (p.34)** to analyze FERA’s behavior when clients use models of substantially different sizes.
  This addresses **Reviewer LBiw – Q2, Q6** and **Reviewer Fjet – Q1**.

- We added **Section F.2: Effect of Uncertainty Characteristics (p.34)** to evaluate FERA’s stability when uncertainty arises purely from decoding stochasticity rather than model capacity.
  This addresses **Reviewer LBiw – Q2**.

- We added **Section H (p.36)** to justify the choice of token-level entropy as the uncertainty measure in FERA, and included **Remark 3.5 (p.6)** to provide a clear rationale for this design choice.
  These revisions address **Reviewer LBiw – Q3**.

- We revised **Remark 3.4 (p.5)** to clarify why uncertainty-aware weighting is necessary and how Eq. (3.3) generalizes the baseline aggregation rule.
  This addresses **Reviewer LBiw – Q4**.

- We added **Remark 3.6 (p.6)** to clarify FERA’s synchronization requirements and position the method within the broader federated systems landscape.
  This addresses **Reviewer Fjet – Q1**.

---

### Meta-Review · Area_Chair_W1DQ · 2025-12-18

**Summary:**

This paper introduces FERA, a training-free federated reasoning framework designed to address privacy and efficiency challenges in distributed large language model environments. By eliminating costly parameter updates, FERA utilizes an iterative server-client collaboration process guided by uncertainty-aware aggregation mechanisms. The framework incorporates uncertainty weighting and self-critique strategies to handle data heterogeneity. Theoretical convergence guarantees and empirical results across multiple benchmarks demonstrate that FERA achieves competitive reasoning performance with significantly reduced communication overhead.

**Strengths:**
1. The paper introduces an innovative training-free federated reasoning framework that effectively balances privacy, computational efficiency, and performance.
2. The framework is supported by a solid mathematical foundation through theoretical convergence analysis using linear self-attention models. Extensive empirical evaluations across multiple reasoning benchmarks show the framework's effectiveness and superiority over traditional baselines.
3. The modular design featuring uncertainty-aware weighting and self-critique mechanisms makes the system readily deployable in real-world federated environments.

**Weaknesses:**
1. The evaluation relies exclusively on general-purpose benchmarks, which may not adequately demonstrate the framework's effectiveness in specialized domains where clients possess unique expertise. The authors should investigate how the system performs when clients have highly specialized knowledge that is absent from pre-trained models.
2. There is a lack of technical detail regarding the computation and calibration of uncertainty estimates across heterogeneous client models. Furthermore, the framework lacks mechanisms to handle adversarial or low-quality clients who might manipulate uncertainty scores to gain disproportionate influence.
3. The robustness of the framework is questioned because reasoning traces from LLMs may be noisy patterns rather than faithful logical processes. Additionally, the server's summarization process might inject its own reasoning bias, potentially distorting the authentic reasoning patterns provided by the clients.
4. It lacks thorough investigation of how varying model capabilities and uncertainty characteristics affect aggregation, especially with mixed model scales or architectures. Comparisons with established federated learning methods are also expected.

**Reviewer Concerns:**

To address Reviewer LBiw’s concerns, the authors have detailed the construction of heterogeneous client datasets across different benchmarks and introduced an experiment to evaluate FERA under domain-imbalanced and specialization-heavy conditions. They have also analyzed FERA’s performance when clients use models of substantially different sizes, and assessed FERA’s stability when uncertainty stems purely from decoding stochasticity rather than model capacity. Additionally, the authors justified the selection of token-level entropy as the uncertainty measure in FERA, and provided a clear rationale for this design choice. The authors have challenged Reviewer R7vv’s evaluation, expressing concerns about the reviewer’s familiarity with the literature on LLM reasoning.

**Reviewer Scores:**

The paper received 3 reviews. Reviewer Fjet gave a positive evaluation but explicitly stated, "I am not an expert in federated learning, so I have given a lower confidence score. Please refer more to the comments from other reviewers. Considering that the other two reviews are more negative and the reviewers are more confident, Fjet is likely to reduce the score.

---

### Decision · Program_Chairs · 2026-01-26

Reject